# Ion identity molecular networking for mass spectrometry-based metabolomics in the GNPS environment

Robin Schmid[1,2,23], Daniel Petras [2,3,4,23], Louis-Félix Nothias[2,23], Mingxun Wang [2], Allegra T. Aron[2], Annika Jagels[5], Hiroshi Tsugawa [6,7,8], Johannes Rainer [9], Mar Garcia-Aloy [9], Kai Dührkop [10], Ansgar Korf [1], Tomáš Pluskal [11], Zdeněk Kameník [12], Alan K. Jarmusch [2], Andrés Mauricio Caraballo-Rodríguez[2], Kelly C. Weldon [2], Melissa Nothias-Esposito[2], Alexander A. Aksenov [2,13], Anelize Bauermeister[2,14], Andrea Albarracin Orio [15], Carlismari O. Grundmann [2,16], Fernando Vargas[2], Irina Koester [3], Julia M. Gauglitz[2], Emily C. Gentry [2], Yannick Hövelmann[5], Svetlana A. Kalinina[5], Matthew A. Pendergraft [3], Morgan Panitchpakdi[2], Richard Tehan[17], Audrey Le Gouellec [18], Gajender Aleti[19], Helena Mannochio Russo[2,20], Birgit Arndt[5], Florian Hübner[5], Heiko Hayen[1], Hui Zhi[21], Manuela Raffatellu [21,22], Kimberly A. Prather[3], Lihini I. Aluwihare[3], Sebastian Böcker [10], Kerry L. McPhail [17], Hans-Ulrich Humpf [5], Uwe Karst[1] & Pieter C. Dorrestein [2,13✉]

Molecular networking connects mass spectra of molecules based on the similarity of their fragmentation patterns. However, during ionization, molecules commonly form multiple ion species with different fragmentation behavior. As a result, the fragmentation spectra of these ion species often remain unconnected in tandem mass spectrometry-based molecular networks, leading to redundant and disconnected sub-networks of the same compound classes. To overcome this bottleneck, we develop Ion Identity Molecular Networking (IIMN) that integrates chromatographic peak shape correlation analysis into molecular networks to connect and collapse different ion species of the same molecule. The new feature relationships improve network connectivity for structurally related molecules, can be used to reveal unknown ion-ligand complexes, enhance annotation within molecular networks, and facilitate the expansion of spectral reference libraries. IIMN is integrated into various open source feature finding tools and the GNPS environment. Moreover, IIMN-based spectral libraries with a broad coverage of ion species are publicly available.

A full list of author affiliations appears at the end of the paper.

Molecular networking (MN)[1] within the GNPS web platform (http://gnps.ucsd.edu)[2] has been used for the analysis of nontargeted mass spectrometry data in various fields[3,4]. MN relies on the principle that similar structures tend to form similar patterns in fragmentation mass spectra (MS[2]). MN is built up through the pairwise spectral comparisons of a dataset, creating an MS[2] spectral network. This network is then enriched by annotating the experimental MS[2] spectra against MS[2] spectral libraries[2,5] or compound databases (Fig. 1). In the resulting molecular networks, annotations can be propagated through the network edges to adjacent nodes[6]. MN can be used to map the chemical space of complex samples to facilitate the discovery of new molecules, especially analogs of known compounds[2]. For the analysis of liquid chromatography-mass spectrometry (LC-MS[2]) data, feature-based molecular networking (FBMN) combines MN with chromatographic feature-finding tools[7].

During LC-MS ionization, a given compound can generate multiple ion species (e.g., protonated and sodiated adducts), which appear as individual nodes in a molecular network, due to different precursor mass-to-charge ratios ($m/z$). As various commonly detected ion adducts exhibit different fragmentation behavior during collisional activation (e.g., in collision-induced dissociation (CID) mode) (Supplementary Fig. 1), MS[2] spectral networking on its own does not necessarily connect all ion adducts produced by a single compound. This often contributes to the unwanted separation of molecular families (subnetworks) and limits the propagation of library annotations through the networks. The two ion species that are most frequently represented in spectral libraries ([M + H]$^+$ and [M + Na]$^+$) typically stay unconnected.

Various tools have been developed for the grouping and annotation of ion species in LC-MS data. The first step, feature grouping, typically involves a retention time filter and the correlation of feature intensities across samples[10–12]. Other tools, such as CAMERA and CliqueMS, add a pairwise correlation of feature shapes to the grouping metric[13,14]. RAMClust provides an option to simultaneously process LC-MS data with MS[2] from data-independent acquisition (DIA)[10]. While many tools[10,12–15] directly interoperate with the feature-finding software XCMS[16], MS-FLO was developed to process exported feature lists from MZmine[17], MS-DIAL[18], and XCMS. Generally, after feature grouping, ion species can be identified based on known mass differences. Connecting all ions that originate from the same molecule results in MS[1]-based groups, here referred to as ion identity networks (IIN).

In this work, we present Ion Identity Molecular Networking (IIMN) and showcase how to fuse MS[2]-based spectral networks with an additional networking layer based on MS[1] feature shape correlation of identified ion species that originate from the same molecule. IIMN addresses this central bottleneck of unconnected ion adducts in MN and the general problem of feature redundancy in MS-based metabolomics[8,9]. We further show the initial validation of IIMN with a ground truth dataset with induced adduct formation by post-column infusion of salt solutions. Furthermore, we present IIMN results for two datasets of natural products standards as well as 24 publicly available experimental datasets.

## Results

**Workflow development.** The IIMN workflow annotates and connects related ion species in feature-based molecular networks within the GNPS web platform. We integrate IIN into MS[2]-based molecular networks and demonstrate the application to LC-MS[2]

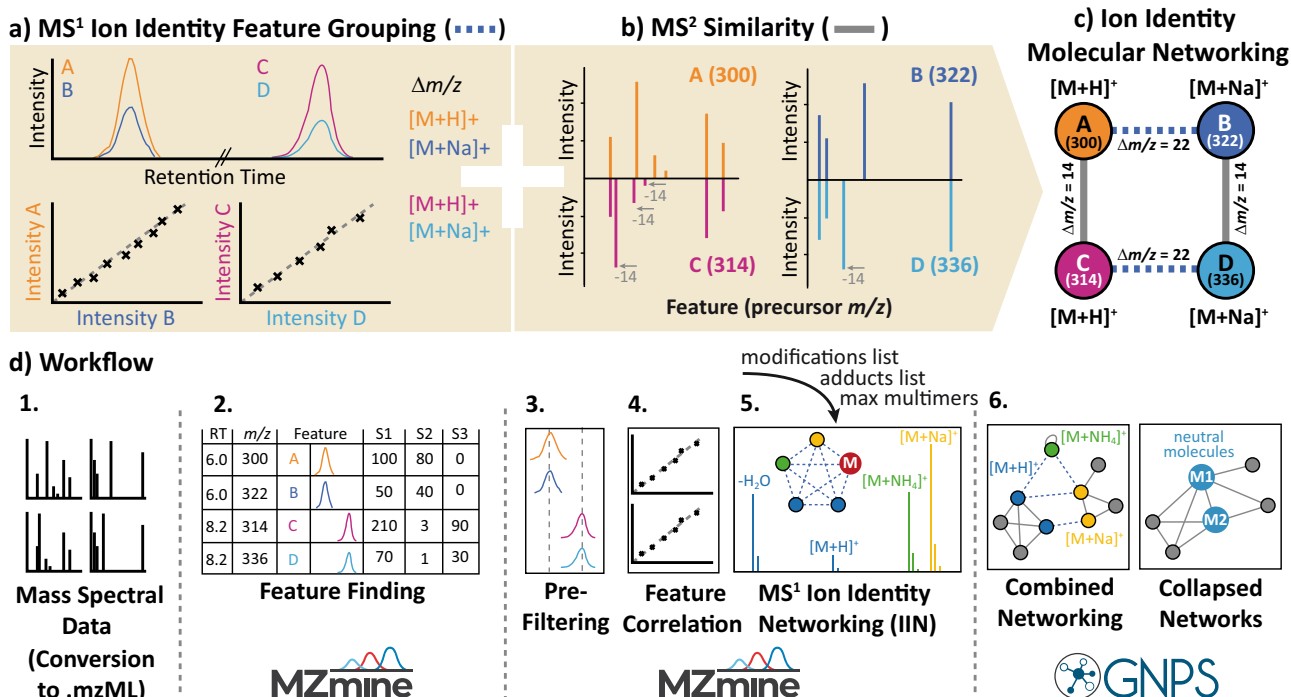

**Fig. 1 The concept of ion identity molecular networking (IIMN).** The workflow integrates **a** MS[1] feature grouping to connect different ion species of the same compound and **b** feature-based molecular networking to connect similar compound structures based on MS[2] spectral similarity to yield **c** combined networks. **d** highlights the data processing steps to create IIMN networks in MZmine and GNPS. After feature detection and alignment across multiple samples, features are grouped based on the correlation of their chromatographic feature shapes (intensity profiles) and other MS[1] characteristics. Subsequently, ion species of grouped features are identified with an ion identity library generated based on user input for included adducts, in-source modifications, and a maximum multimer parameter. After uploading these results to the GNPS web server, the IIMN workflow generates combined networks and an alternative output with all IIN collapsed into single molecular nodes to reduce complexity and redundancy.

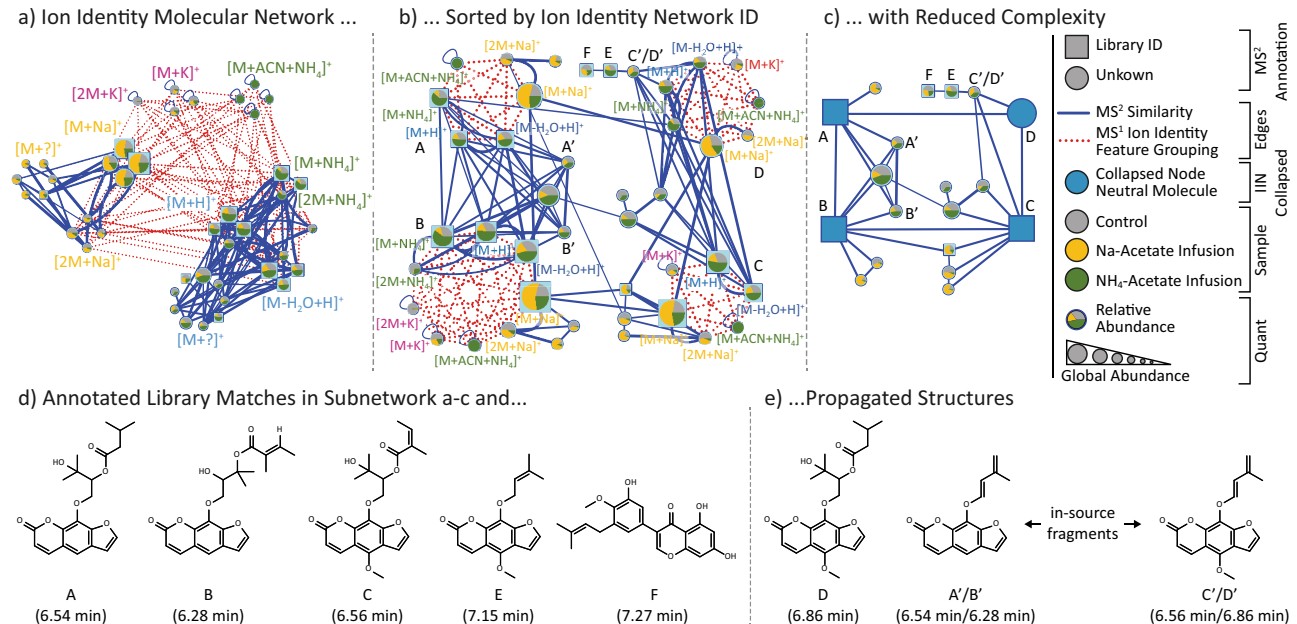

**Fig. 2 Ion identity molecular networking.** Depicted are three visualizations of the same ion identity molecular network from the post-column salt infusion experiments. **a** Sorting by ion identities reveals that $MS^2$ similarity edges (blue) often link sodiated ions (e.g., $[M + Na]^+$ and $[2M + Na]^+$) into a subnetwork that is separated from a subnetwork of ammonium adducts with protonated species. The pie charts indicate relative abundances in different salt addition experiments (Control ($H_2O$), gray; Na-Acetate, yellow; $NH_4$-Acetate, green). The complexity and redundancy are reduced by **b** sorting all ions of the same molecule in a circular layout and **c** collapsing all IIN into representative single molecular nodes. This option reduces the complexity of this IIMN from 43 feature nodes to four molecular nodes (A–D) and 15 feature nodes (−56%). **d** Lists the structure of all GNPS library matches and **e** propagated structures for D (based on A and C) and the in-source fragments A′ to D′. This subset of structurally related compounds gives a first statistical proof for high correct annotation rates during IIN in MZmine as adduct formation responds to the corresponding salt infusion, e.g., higher $[M + Na]^+$ abundances in the sodium acetate buffer infusion.

studies that make use of product ion scans acquired in data-dependent acquisition (DDA) mode. The IIMN workflow comprises feature grouping, feature shape correlation, and identification of ion species using a variety of feature-finding software tools, such as MZmine[17], XCMS[16], and MS-DIAL[18] that make use of different algorithms for the identification of ion adducts. A table of extracted $MS^1$ features, each with a consensus $MS^2$ spectrum, together with IIN results are then uploaded to GNPS to run the IIMN workflow on the web server. The resulting ion identity molecular networks contain two layers of feature (node) connectivity, linking ion identities of the same compound by $MS^1$ characteristics and structurally similar compounds by $MS^2$ spectral similarity (Fig. 1). A detailed description of the IIMN workflow as well as a step-by-step tutorial are provided in the method section and can be found online in the GNPS documentation (https://ccms-ucsd.github.io/GNPSDocumentation/fbmn-iin/). The IIMN workflow is available online (https://gnps.ucsd.edu/) and the source code is shared on GitHub under an open source license.

**Validation of IIMN by post-column infusion of salt solutions.** To validate the identification of ion species with IIMN, we created an LC-$MS^2$ benchmark dataset of a natural product mixture containing 300 compounds, in which we promoted adduct formation by post-column infusion of ammonium acetate or sodium acetate at different concentrations (Fig. 2a–e). The IIMN networks can be depicted in alternative layouts that illustrate complementary results within the same dataset. GNPS also provides networks with collapsed IIN to reduce the redundancy of different ion species by merging them into a single neutral molecule (M) node (Fig. 2c). In this dataset, IIMN successfully connects ion identities and reduces the size of a complex network by 56% to

four major compounds. The increased connectivity facilitates the propagation of structure annotations to neighboring in-source fragments and an unannotated compound. Finally, the abundance change of identified adducts ($[M + H]^+$, $[M + NH_4]^+$, $[M + Na]^+$) in our benchmark dataset is in agreement with the different post-chromatography salt infusion conditions ($H_2O$, Na-Acetate, or $NH_4$-Acetate, Fig. 3), which validates ion species identification on a dataset level. For instance, the abundance of $[M + Na]^+$ and $[M + NH_4]^+$ ion identities was significantly ($p < 0.001$) higher in the corresponding samples with the post-column infusion of sodium acetate or ammonium acetate, respectively, when compared to the control samples. The exclusive formation of an uncommon $[M + ACN + NH_4]^+$ in-source cluster after infusion of ammonium ions into an ACN-water gradient further verifies connected ion identities.

**Application of IIMN to 24 public datasets.** To test the workflow with data generated from various sample types and on different experimental platforms, 24 public datasets were processed using the MZmine workflow and its metaCorrelate algorithm for feature grouping and ion identity networking (Fig. 4, Supplementary Table 1). All the specific parameters for processing are provided in the methods section (under Dataset processing). MZmine feature-finding parameters were optimized for each dataset by various coauthors, while the feature grouping and ion identity networking parameters were kept constant for better comparability. IIMN identified biologically relevant metal-binding compounds via post-column-induced ion species. In a native ESI-based metabolomics study, IIMN specifically revealed that the known siderophore yersiniabactin also acts as a zincophore (Supplementary Note 1, Supplementary Fig. 2)[19] and was validated in animal experiments.

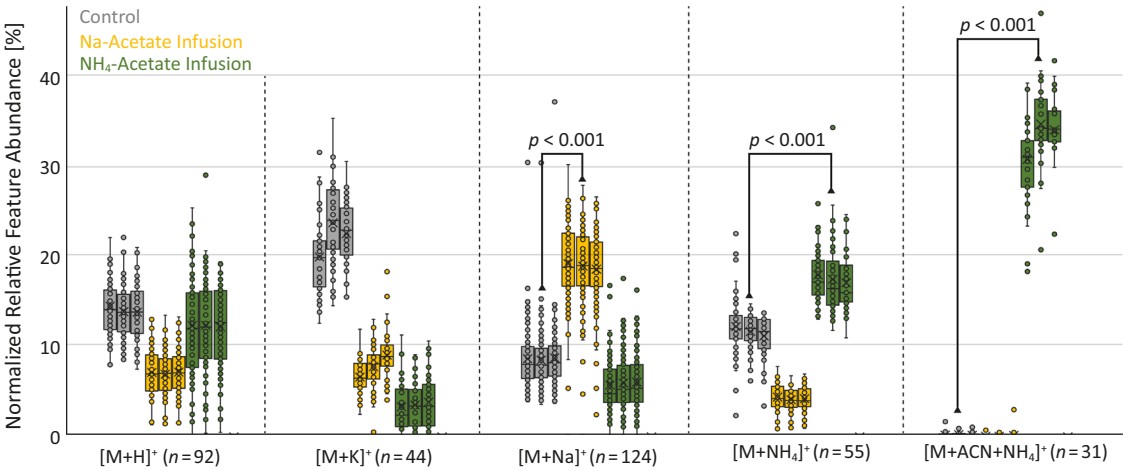

**Fig. 3 Statistical impact of salt addition experiments on ion identity abundances.** The relative intensities of selected ion identities are plotted for each post-column infusion in triplicate. The significant change for $[M + Na]^+$ and $[M + NH_4]^+$ ion identities in the corresponding post-column salt infusions compared to the control samples agree with the expected ionization behavior. The exclusive formation of an uncommon $[M + ACN + NH_4]^+$ in-source cluster in the ammonium acetate buffer infusion further verifies ion identity networking results. Boxplots visualize the median as a horizontal line, the mean as an x, the first (Q1) and third quartile (Q3) as the lower and upper hinges, and the whiskers corresponding to the minimum value below Q1 and the maximum value above Q3 within the $1.5 \times IQR$ (where IQR is the interquartile range). The $p$-values of a Welch two-samples $t$-test and the corresponding number of ion identities n are provided for each pair of compared triplicate injections with different post-column salt infusion conditions. Source data are provided as a Source Data file.

For a dataset with 88 extracts from feces and gall bladder of various animals, the comparison between feature-based molecular networking with and without the additional edges from ion identity networking demonstrates how IIMN complements and improves FBMN (Fig. 5). Here, IIMN combined multiple smaller networks and unconnected nodes into a large network of free bile acids and those conjugated to amino acids or sulfate. These results prove that IIMN is a suitable method to connect structurally similar compounds, such as isomers, based on $MS^2$ spectral similarity scoring and feature shape correlation. FBMN only established one edge between subnetworks of free and conjugated bile acids. Overall, bile acid analogs were separated into multiple subnetworks and unconnected nodes with a clear trend of separating sodiated and protonated ion identities. Finally, the complexity and redundancy are reduced by collapsing all IINs into corresponding representative nodes. The final network has a reduced number of nodes and a higher density of edges between nodes with annotations to the same compound classes.

IIMN also yielded additional structural information in the case of samples from the mold *Stachybotrys chartarum* (Supplementary Note 2, Supplementary Fig. 3). In this project, IIMN revealed novel phenylspirodrimane derivatives, which were verified by nuclear magnetic resonance spectroscopy (NMR)[20]. In the network, the increasing number of aliphatic hydroxyl groups was reflected by the maximum number of in-source water losses, whereas acetylation of hydroxy groups reduced this number. The manual inspection of IIMN results was facilitated by additional $MS^1$ annotations provided by ion nodes that lack $MS^2$ fragmentation data and are consequently unavailable to the FBMN workflow. During the creation of IIMN networks, further layers of additional feature connections can be supplied. One example is a relationship between ion identity networks based on neutral mass differences that annotate putative structure modifications between compounds (Supplementary Note 3, Supplementary Fig. 4).

From a global view on all 24 datasets, IIMN successfully reduced the number of unconnected LC-$MS^2$ features and increased the connections to annotated compound structures

(Supplementary Fig. 5, Supplementary Table 2). Annotation rates in all 24 datasets of 6% and 12% are in the expected range with contemporary $MS^2$ library matching[21,22] and $MS^1$ ion annotation, respectively, especially with the here chosen restrictive IIN parameters (Fig. 4a). By propagating spectral library matches to first neighboring IIN nodes, the annotation rates of the test datasets were increased by an average of 35% (Fig. 4a, b). On the individual dataset level, the highest increase (325%) was observed for dataset 4 with more $MS^1$ data points per feature and thus better feature shape correlation on the cost of a lower $MS^2$ acquisition rate. Most datasets (16 out of 24) experienced an increase greater than 10%, while eight datasets were below this value. After inspecting the LC-$MS^2$ files, we found various reasons for this difference. Datasets 11 and 12, for example, had a higher focus on $MS^2$ acquisition with a high topN of $MS^2$ events in the DDA settings that caused lower $MS^1$ survey scan frequencies and hence fewer data points per features, resulting in lower IIN correlation and connectivity. For datasets 7 and 19, the $MS^2$ annotation rate was low to begin with and hence few annotations could be propagated by IIMN.

**Generation of IIMN-based spectral libraries**. Besides the increase in feature annotations in individual datasets, IIMN also enables the generation of propagated spectral libraries, increasing and diversifying the library coverage beyond commonly considered ion species. In positive ion mode, for example, most mass spectrometrists routinely consider $[M + H]^+$ and $[M + Na]^+$ adducts, but less frequently $[M + NH_4]^+$, $[M + Ca]^{2+}$, $[M + K]^+$, and in-source fragments in their data analysis and hence library contributions. However, while inspecting the relative distribution of ion identities within all 24 datasets, marine samples showed a higher percentage of $[M + NH_4]^+$ adducts ($24 \pm 5\%$) when compared to all other datasets ($10 \pm 8\%$). Sodiated adducts that were expected to be elevated in marine samples (due to anticipated higher salt contents in the original sample), in contrast, are evenly distributed between all datasets with an average of $26 \pm 6\%$ (Fig. 4c). On average, protonated species contribute to $23 \pm 6\%$ of the overall ion identities in our test

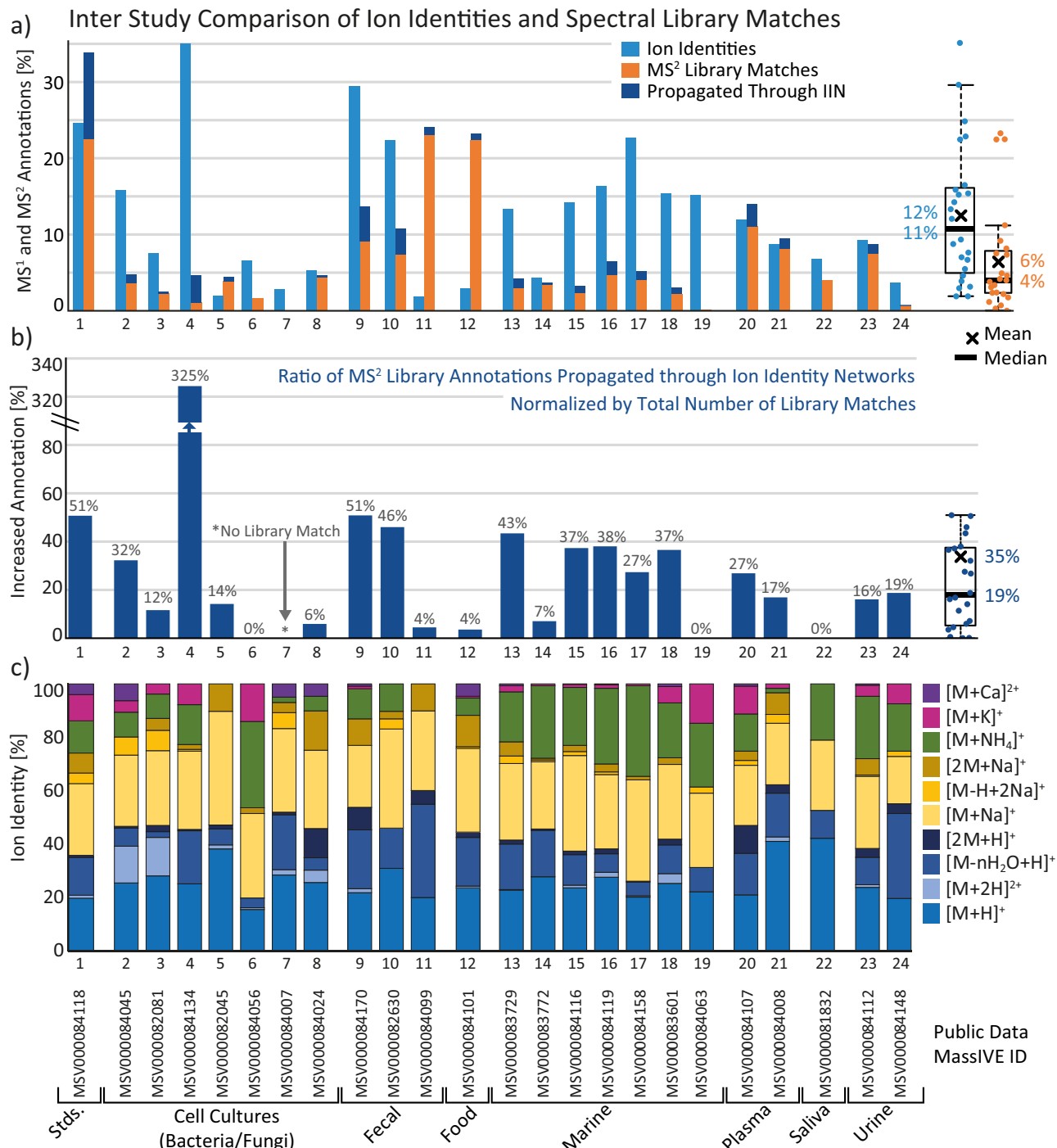

**Fig. 4 Overview of IIMN results for 24 experimental datasets. a** Summarizes the relative number of LC-MS features (with an MS² spectrum) that were annotated by ion identities or matches to the GNPS spectral libraries. The increased annotation rate by propagating library matches to connected unannotated ion identities is highlighted and **b** displayed as relative gains with a mean increase by 35% compared to all library matches. **c** Comparison of relative ion formation tendencies measured as the number of ion identities. Boxplots summarize the statistics of overall $n = 24$ datasets by visualizing the median as a horizontal line, the mean as an x, the first and third quartile as the lower and upper hinges, and the whiskers corresponding to the minimum value below Q1 and the maximum value above Q3 within the 1.5 × IQR. Source data are provided as a Source Data file.

datasets, indicating spectral bias in public MS² libraries such as MassBank of North America (66% $[M + H]^+$) and GNPS (65% $[M + H]^+$) (Fig. 6), and suggests that the community should provide MS² spectra for other ion species of the same molecules to reference libraries. Here, IIMN can be used to expand the spectral libraries with additional adducts and in-source fragments in LC-MS experiments, which can significantly increase spectral

library coverage and thus MS² annotation rates. The potential to use IIMN to propagate spectral library matches to adjacent unannotated features with ion identity is evident from a mean increase of the annotation rate by 35% (Fig. 4a, b). By propagating high confident spectral matches (in this case, cosine >0.9 or authentic standards) to connected ion identities from the 24 public datasets and two datasets of natural products from the

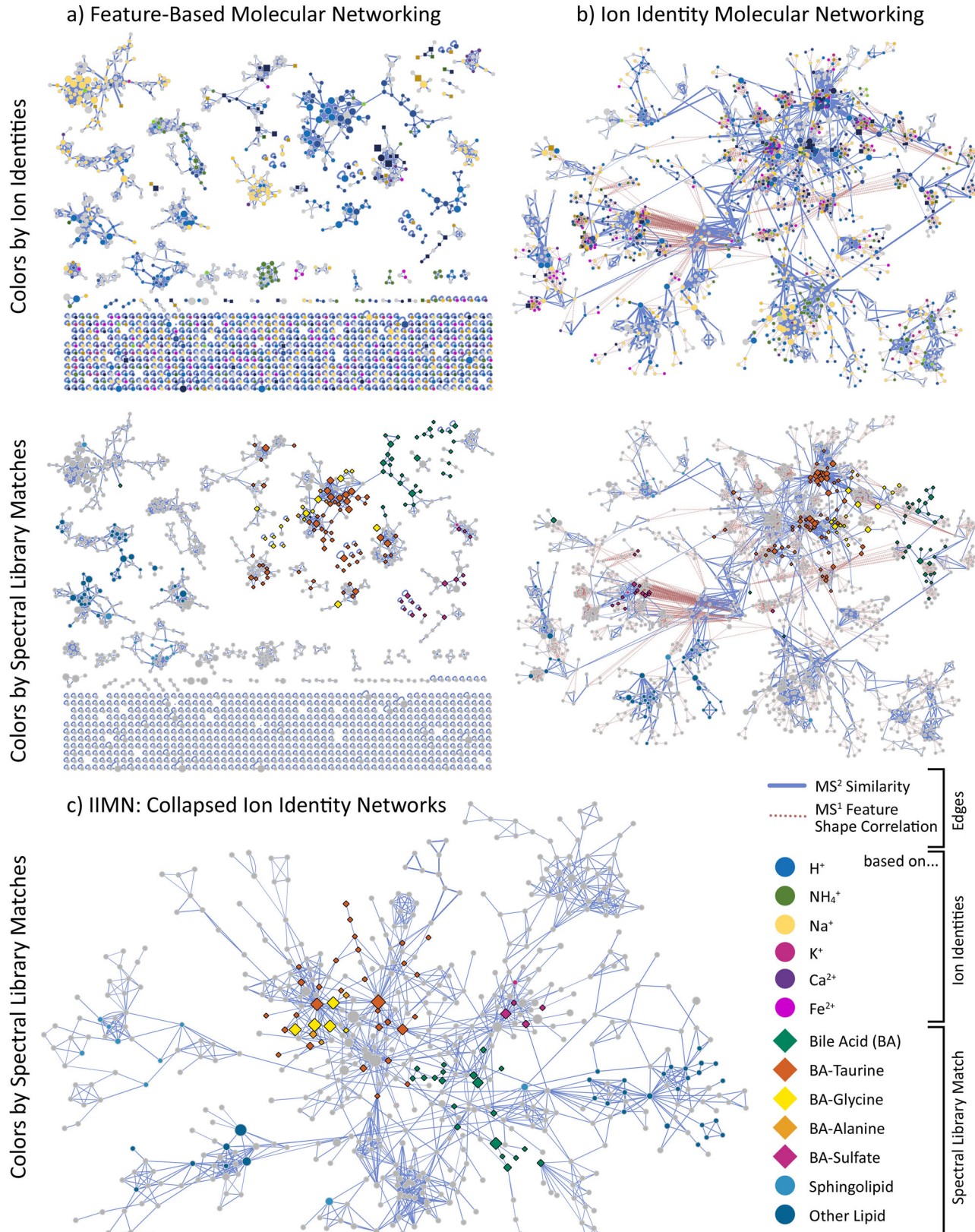

**Fig. 5 Comparisons of a subnetwork with matches to bile acids from 88 feces and gall bladder samples of various animals (MSV000084170).** This overview compares **a** the FBMN results to IIMN **b** before and **c** after collapsing all ion identity networks into single representative nodes. In the top row, nodes are colorized depending on the adduct that ion identities are based on. In contrast, the lower three networks emphasize nodes with MS$^2$ spectra that match library spectra of specific compound classes, mainly bile acids and their conjugates. The collapsed network (**c**) reduces the complexity and redundancy of having multiple nodes per compound and only keeps MS$^2$ spectral similarity edges.

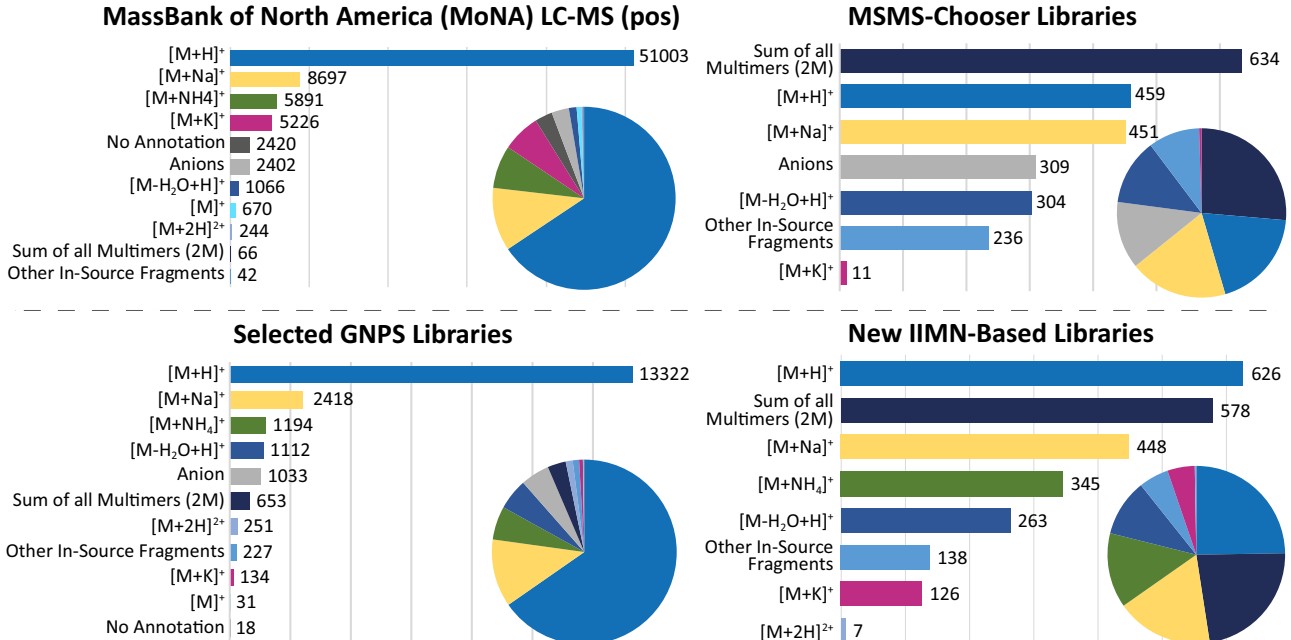

**Fig. 6 Analysis of the coverage and distribution of ion identities in public LC-MS$^2$ spectral libraries (refer to Supplementary Table 3 for library origins).** Two-thirds of the MassBank of North America LC-MS$^2$-positive ion mode library entries were entered as $[M + H]^+$ while only four other ion types reached more than 1000 entries, namely, $[M + Na]^+$, $[M + NH_4]^+$, $[M + K]^+$, and $[M − H_2O + H]^+$. Other in-source fragments, multiply charged species, and multimers are only covered for a few compounds. A significant number of entries were either annotated as negatively charged adducts (e.g., $[M − H]^−$) or were missing an annotation. As the ion identity naming was not harmonized, different versions pointing to the same ion identity were added to a total count. A similar ion annotation coverage was found in the GNPS spectral libraries. In contrast, libraries that were generated with the recently described MSMS-Chooser[29] workflow on GNPS or the IIMN-based library extraction workflow, described here, show an overall broader coverage of different adducts, multimers, and in-source fragments. The depicted statistical visualization compares a subset of significant or representative ion identities. The IIMN-based numbers summarize the libraries from both the 24 experimental datasets and the two NIH natural product standards datasets with a total of 2659 library entries. Source data are provided as a Source Data file.

NIH 'ACONN' collection from which an original reference library was created, we created IIMN spectral libraries with a total of 2657 entries with a broader and more representative ion species coverage (e.g., 24% $[M + H]^+$, 22% multimeric species, 17% $[M + Na]^+$, 15% in-source fragments, and 13% $[M + NH_4]^+$). Such spectral libraries better represent ion species observed in typical metabolomics experiments (Supplementary Table 3 and Fig. 6).

## Discussion

In conclusion, by establishing relationships between different ion species originating from the same compound and structurally similar compounds, IIMN facilitates molecular network interpretation and compound annotation. The combined networks with two layers of feature connectivity enable strategies to present and analyze mass spectrometry data. Networks with collapsed IIN especially reduce the redundancy of detecting multiple ion species per compound. IIMN successfully connected more related compound annotations in datasets from different analytical platforms and over a variety of small molecule compound classes, hence reducing the number of unconnected nodes and increasing the annotation density in molecular networks. An exciting application of IIMN is the expansion of spectral libraries by (re)-processing public datasets and propagating spectral library annotations to create library entries of connected ion identities. The identification of ion adducts can reveal novel ionophores, some of which will be biologically relevant and are still underappreciated in the function of small molecules[19,23]. The integration into FBMN and the GNPS environment provided a platform to utilize IIMN in other related bioinformatics tools, e.g.,

SIRIUS[24], CANOPUS[25], and Qemistree[26] for molecular formula and compound class level annotation, which will also facilitate additional validation of network connectivity. Direct interfaces to the GNPS-Dashboard and MASST[27] support collaborative data visualization and repository scale MS$^2$ queries, respectively. Furthermore, the open source code and generic connection between feature finding, ion identity molecular networking, and the online GNPS workflow encourage the implementation of interfaces to other feature grouping and ion identification algorithms. We anticipate that the option to add orthogonal relationships between features to IIMN will stimulate the integration and development of additional tools for spectral alignment and measures of feature–feature relationships[28].

To reach a broad user base, we interfaced the IIMN workflow with three widely used open source MS processing tools (MZmine[17], MS-DIAL[18], and XCMS[13,16]). Detailed documentation and training videos are available online (https://ccms-ucsd.github.io/GNPSDocumentation/fbmn-iin/). The option to directly submit IIMN analysis from MZmine to GNPS in particular provides a simple entry point for new users.

## Methods

**Post-column salt infusion experiments.** For salt addition UHPLC-MS$^2$ experiments, a mixture of 300 natural products from the NIH NCGC collection was prepared in 100 μL methanol-water-formic acid (80:19:1, Fisher Scientific, San Diego, USA) at a concentration of 0.01 μM of which 2 μL were injected into a Vanquish UHPLC system coupled to a Q-Exactive quadrupole orbitrap mass spectrometer (Thermo Fisher Scientific, Bremen, Germany) in three technical replicates. For the chromatographic separation, a reversed-phase C18 porous core-shell column (Kinetex C18, 50 × 2 mm, 1.8 um particle size, 100 Å pore size, Phenomenex, Torrance, USA) was used. For gradient elution, a Vanquish (Thermo Fisher Scientific, Bremen, Germany) high-pressure binary gradient system was used. The mobile phase consisted of solvent A $H_2O$ + 0.1% formic acid (FA) and

solvent B acetonitrile (ACN) + 0.1% FA. The flow rate was set to 0.5 mL/min. Samples were eluted with a linear gradient from 0–0.5 min, 5% B, 0.5–8 min 5–50% B, 8–10 min 50–99% B, followed by a 2 min washout phase at 99% B and a 3 min re-equilibration phase at 5% B. Post-column we infused ammonium acetate or sodium acetate solutions (50, 5 and 0 mg/L) at 10 µL/min (dilution factor 50) with a syringe pump to yield a final concentration of sodium or ammonium acetate of 1, 0.1, and 0 mg/L. Data-dependent acquisition (DDA) of $MS^2$ spectra was performed in positive mode. Electrospray ionization (ESI) parameters were set to 52 psi sheath gas pressure, 14 AU auxiliary gas flow, 0 AU sweep gas flow, and 400 °C auxiliary gas temperature. The spray voltage was set to 3.5 kV and the inlet capillary to 320 °C. 50 V S-lens level was applied. MS scan range was set to $m/z$ 150–1500 with a resolution at $m/z$ 200 of 17,500 with one micro-scan. The maximum ion injection time was set to 100 ms with an automatic gain control (AGC) target of 1E6. Up to 5 $MS^2$ spectra per $MS^1$ survey scan were recorded in DDA mode with a resolution of 17,500 at $m/z$ 200 with one micro-scan. The maximum ion injection time for $MS^2$ scans was set to 100 ms with an AGC target of 3.0E5 ions and a minimum 5% C-trap filling. The $MS^2$ precursor isolation window was set to $m/z$ 1. The normalized collision energy was set to a stepwise increase from 20 to 30 to 40% with a single charge as the default charge state. $MS^2$ scans were triggered at the apex of chromatographic peaks within 2–15 s from their first occurrence. Dynamic precursor exclusion was set to 5 s. Ions with unassigned charge states were excluded from $MS^2$ acquisition as well as isotope peaks.

**Ion identity molecular networking-workflow overview**. In general, the ion identity molecular networking (IIMN) workflow starts with LC-$MS^2$ data processing in one of the supported open source feature-finding tools. After the creation of an aligned feature list of all samples, ion species that originate from the same analyte are grouped and annotated by $MS^1$ criteria, such as their retention time, feature shape correlation, and $m/z$ difference. Here, such groups are named ion identity networks. Subsequently, information of detected features and their representative $MS^2$ spectra, ion identities, and connections to other ion identities are exported and transferred to the GNPS web server for the molecular networking part (refer to tool-specific sections for details). After the construction of ion identity molecular networks, features share connectivity based on $MS^2$ spectral cosine similarity and $MS^1$-based feature shape correlation. In addition to this combined network, GNPS calculates a version with collapsed IIN, where one node represents multiple ions of the same molecule. Results are available in the GNPS web interfaces and as downloads in various open formats as tables and networking files to allow local visualization, reviewing, and post-processing.

The IIMN workflow aids the feature-based molecular networking workflow[7] by adding $MS^1$ specific information, which is provided as new columns in the quantification table and as additional edges in a Supplementary Pairs text file within the GNPS-FBMN workflow. The option of additional edges from other tools was introduced to stimulate and facilitate the development of new computational methods that link nodes in the resulting molecular networks and was initially developed for IIMN. The text format follows a generic comma-separated style with the columns ID1 and ID2 (matching the feature IDs in the feature quantification table and mgf), EdgeType (defining the method), Score (numerical), and Annotation. To enable a broad user base to employ ion identity molecular networking in their studies, three popular mass spectrometry processing tools, namely, MZmine[17], MS-DIAL[30], and XCMS(+ CAMERA)[13,16], were modified or adapted with additional export scripts or modules. In comparison to FBMN, IIMN can include features that are lacking $MS^2$ fragmentation spectra but are connected to other feature nodes by $MS^1$ IIN edges. Regarding a higher detectability by $MS^1$ compared to triggered $MS^2$ acquisition, the additional nodes with ion identities complement the resulting networks with information otherwise lost in FBMN or classical MN.

*The general steps to create ion identity molecular networks.*

(1) If needed, convert the spectral data files to an open format (e.g., mzML)
(2) Import the data into one of the open source tools: MZmine, MS-DIAL, or XCMS
(3) Process the data to create a feature list (aligned overall samples)
(4) Perform $MS^1$-based feature grouping and ion identity annotation
(5) Export the feature list as a feature quantification table (.csv), an $MS^2$ spectral summary file (.mgf), which contains a representative fragmentation spectrum for each feature, and a supplementary edges files (IIN files, .csv) (more information in the tool-specific workflow sections)
(6) Create a metadata file to group samples for statistics (optional)
(7) Upload all files to GNPS and start a new feature-based molecular networking job (MZmine can directly submit and start a new IIMN job on GNPS)
(8) Download and visualize the results in a network analysis software (e.g., Cytoscape[31], https://cytoscape.org/)
(9) The option Download Cytoscape Data provides two.graphml networking files

 (a) The standard FBMN and IIMN networks (base directory)
 (b) IIMN networks with collapsed ion identity networks (in the gnps_molecular_network_iin_collapse_graphml directory)

(10) The option Direct Cytoscape Preview/Download provides the IIMN network and its collapsed version as Cytoscape projects with various style presets

Refer to the documentation on how to run FBMN within GNPS and multiple mass spectrometry data processing tools.

https://ccms-ucsd.github.io/GNPSDocumentation/featurebasedmolecularnetworking/

For IIMN, refer to the related part of the GNPS documentation.

https://ccms-ucsd.github.io/GNPSDocumentation/fbmn-iin/

*Generation of collapsed ion identity networks*. One result of the GNPS-IIMN workflow is the combined networks with IIN collapsed into single nodes. For this, all ion nodes with the same IIN ID are merged into a representative node based on the feature with the highest library match score, if available, or otherwise the feature with the maximum abundance. While all IIN edges are collapsed, MN edges of all ion identities are redirected to their representative nodes so that duplicates replace existing edges if their edge score (cosine similarity) is higher. Limiting the number of MN edges to the one with the highest cosine similarity. Furthermore, representative collapsed nodes are extended by multiple attributes, including the intensity of each ion identity and their summed intensity. This enables the direct comparison of ionization tendencies and provides new visualization options. An example with pie charts of the ion abundances is demonstrated in Supplementary Fig. 3.

*Cross-validation of $MS^2$ spectral library matches and $MS^1$ ion identities*. In IIMN, nodes may combine annotations from $MS^2$ spectral library matching and $MS^1$ ion identity networking. As cross-validation, GNPS parses and harmonizes the ion species string of both the detected ion identity and matching spectral library entry before checking for equality. The results are reported as an additional column in the node table. This equality check facilitates manual reviewing and the spotting of discrepancies between the $MS^1$ and $MS^2$ annotations.

The ion string parser harmonizes an input (e.g., $[M - H_2O + 2H]^{2+}$) in the following steps:

(1) Spaces are removed
(2) Charge state is detected and removed from the input (2+)
(3) Brackets are removed ([]())
(4) Input is split into added (+2H) and removed (−H₂O) parts
(5) Both lists are sorted alphabetically (+2H sorted by letter H)
(6) If the charge state is missing, it is calculated for all parts that are listed in a lookup table (e.g., +Na or +H correspond to charge 1+)
(7) The harmonized string is constructed by concatenation of [M-all removed parts + all added parts]charge state.

As an example, the harmonized string $[M + H]^+$ is produced by the input strings M + H, M + H +, and $[M + H]^+$, which are all commonly found in the GNPS spectral libraries and as an output of various software tools.

The full open source code of the ion string parser and its latest charge lookup table can be found on GitHub (https://github.com/CCMS-UCSD/GNPS_Workflows).

**IIMN with MZmine**. MZmine lacked a functional algorithm to group and annotate different ion species of the same molecules. Therefore, a workflow was implemented and split into separate modules for feature grouping (metaCorrelate), annotation of the most common ions (ion identity networking), an option to add more ion identities to existing IIN iteratively, and modules to validate multimers and in-source fragments based on $MS^2$ scans. Both the creation and expansion of ion identity networks follow customizable lists of adducts and in-source modifications to cover any type of multimers, in-source fragments, and adducts. The IIN procedure lists all possible ion identity pairs between two features and ranks them according to the maximum number of features that support a specific annotation, i.e., the ion identity network size. While a feature might be annotated as two different ion species that point to different metabolites, the current workflow will only create additional IIN edges and ion species metadata for the highest-ranking ion identity per feature. This filter decreases the number of spurious matches. Finally, the GNPS-FBMN export module was modified to export all needed files to run IIMN. The quant table (.csv) contains grouping and ion identity specific columns, and a new Supplementary Pairs text file lists all additional IIN edges. The user can limit the export to features with $MS^2$ fragmentation spectra or include those with an ion identity. Consequently, the IIMN workflow on GNPS connects features without $MS^2$ spectra only by their IIN edges. MZmine is the first tool to provide a direct submission to GNPS to start analysis jobs, consequently streamlining the workflow and lowering the entrancing energy needed to apply IIMN within GNPS.

In detail, the metaCorrelate feature grouping algorithm searches for features with similar average retention times, chromatographic intensity profiles (feature shapes) with a minimum percentage of intra-sample correlation and overlap, and minimum feature intensity correlations across all samples (Supplementary Fig. 6). The feature shape correlation is a vital filter to reduce false grouping significantly and can apply either a minimum Pearson correlation (favored) or cosine similarity.

A requirement is at least five data points, two on each side of the peak apex. If a low $MS^1$ scan rate leads to chromatographic peaks with less than five data points, it is advisable to either redesign the acquisition method or to turn off the feature shape correlation. Note that the latter is expected to reduce the ion annotation consistency and should be used with caution. Similarly, the feature height correlation across all samples is optional, provides the same correlation or similarity measures, and additionally, relies on constant ionization conditions for all samples. Therefore, this filter should be turned off if the conditions were changed throughout the study, e.g., by changing the separation conditions or ion source parameters. The general principle of the feature height correlation is that different ions of the same molecule should follow a similar trend in abundance across all samples of the same study. If any feature, such as an $[M + H]^+$ feature, increases at least 10-fold, all grouped features, e.g., $[M + Na]^+$ or $[M + NH_4]^+$, should never have a negative feature height correlation coefficient and should as well increase in abundance. If both the feature shape and feature height correlation filters are omitted, feature grouping is solely filtered by the retention time window and overlap. To annotate features on an $MS^1$ level, ion identity libraries are created with a user-defined list of in-source modifications (fragments and clusters), a list of adducts, and a maximum multimers number parameters (Supplementary Fig. 6). Each adduct is combined with each modification to fill the library with ion identities for 1 M to the maximum multimers number. Ion identity networks are then created by applying all ion identity pairs to all pairs of grouped features to calculate and compare the neutral masses of features with specific ion identities (mass difference, charge ($z$), and multimer number). Optionally, after the creation of ion identity networks with the main library, further ion identities can be added iteratively to existing networks. This workflow enables the user to divide into commonly and uncommonly detected ion identities and ensures that each network contains at least two or more main identities. Finally, an ion identity network refinement provides filters for minimum network size and to only keep the largest (most descriptive) IIN per feature.

More on the integration of the new IIMN workflow in MZmine can be found online (http://mzmine.github.io/iin_fbmn).

Refer to the documentation and video tutorials on how to apply IIMN within MZmine and GNPS. The Youtube playlist "MZmine: Ion Identity Molecular Networking" contains instructions on data processing for IIMN and FBMN, a minimalistic and full IIMN workflow within MZmine, and theoretical background to feature shape correlation and ion identity molecular networking.

https://ccms-ucsd.github.io/GNPSDocumentation/fbmn-iin-mzmine/
https://www.youtube.com/playlist?list=PL4L2Xw5k8ITyxSyBdrcv70LDKsP8QNuyN

**IIMN with XCMS (CAMERA).** The XCMS[16] Bioconductor package[32] is the most widely used software for processing untargeted LC-MS-based metabolomics data. Its results can be further processed with the CAMERA[13] package to determine which of the extracted $m/z$-rt features might be adducts[13] or isotopes[33] of the same original compound. For the integration of XCMS and CAMERA into the IIMN workflow, utility functions were created ('*getFeatureAnnotations*' and '*getEdgelist*') to extract and export $MS^1$ based feature and edge annotations (i.e., grouping of features to adduct/isotope groups of the same compound). In addition, the utility function '*formatSpectraForGNPS*' is used to export $MS^2$ spectra. These functions are available in the GitHub repository https://github.com/jorainer/xcms-gnps-tools. R-markdown documents and python scripts with example analyses and descriptions are available in the documentation. (https://ccms-ucsd.github.io/GNPSDocumentation/fbmn-iin-xcms/) The files exported by these utility functions can be directly used for IIMN analysis on GNPS. Note that theoretically, it is possible to use RAMClust[10], CliqueMS[14], or other packages available for XCMS that perform ion annotation. The results of these packages need to be reformatted to the introduced generic supplementary edges format. The CAMERA integration might serve as a reference and starting point.

**IIMN with MS-DIAL.** MS-DIAL[34] is a polyvalent mass spectrometry data processing software capable of processing various nontargeted LC-MS metabolomics experiments, including ion mobility mass spectrometry (http://prime.psc.riken.jp/compms/msdial/main.html). MS-DIAL supports IIMN since version 4.1. After a standard data processing workflow with MS-DIAL, the alignment results can be exported for IIMN analysis using the GNPS export option. Detailed documentation and representative tutorials are available in the GNPS documentations (https://ccms-ucsd.github.io/GNPSDocumentation/fbmn-iin-msdial).

**Dataset processing.** All 24 datasets (Supplementary Table 1) were processed with the MZmine workflow. As each dataset originates from a different study and was acquired with different LC-MS methods, variable feature detection and alignment parameters were applied, which are summarized in Supplementary Table 4. For all datasets, the same parameters were used for the feature grouping module (meta-Correlate) and the ion identity networking modules, with the only exception that the feature height correlation filter was turned off to group features for the post-column salt infusion experiments. As described previously, this filter should only be applied if the ionization conditions and detection sensitivity are kept constant

overall samples. The post-column infusion of different salt solutions for this study promotes the formation of specific ion species in the ionization source.

(1) A pair of features were grouped with a retention time tolerance of 0.1 min, with a minimum overlapping intensity percentage of 50% in at least two samples in the whole dataset (gap-filled features excluded), a feature shape Pearson correlation greater equals 0.85 with at least five data points and two data points on each edge, and a feature height Pearson correlation greater equals 0.6 with at least three data points.

(2) The initial creation of ion identity networks was performed using the ion identity networking module and a maximum tolerance of 0.001 $m/z$ or 10 ppm, a comparison where a pair of features and a pair of ion identities only need to match in one sample, and an ion identity library created based on 2 M as the maximum multimers number, a list of adducts ($[M + H]^+$, $[M + Na]^+$, $[M + NH_4]^+$, $[M - H + 2Na]^+$, $[M + 2H]^{2+}$, and $[M + H + Na]^{2+}$), and a list of in-source modifications ($[M - H_2O]$ and $[M - 2H_2O]$).

(3) Two iterations were applied to add more ion identities to the resulting networks of step 2 with an unchanged $m/z$ tolerance.

(a) To add a higher variety of adducts, a maximum multimers number of 2, a list of adducts ($[M + H]^+$, $[M + Na]^+$, $[M + K]^+$, $[M + NH_4]^+$, $[M - H + 2Na]^+$, $[M - H + Ca]^+$, $[M - H + Fe]^+$, $[M + 2H]^{2+}$, $[M + H + Na]^{2+}$, $[M + H + NH_4]^{2+}$, $[M + Ca]^{2+}$, and $[M + Fe]^{2+}$), and an empty list of modifications were used.

(b) To add a greater variety of modifications and larger multimers, a maximum multimers number of 5, a list of adducts ($[M + H]^+$, $[M + NH_4]^+$, and $[M + 2H]^{2+}$), and a list of modifications ($[M - H_2O]$, $[M - 2H_2O]$, $[M - 3H_2O]$, $[M - 4H_2O]$, $[M - HFA]$, and $[M - ACN]$) were used.

**Dataset statistics.** Ion identity molecular networking statistics on all datasets were extracted with a new MZmine module and exported to a comma-separated file (csv) for evaluation in Microsoft Excel. The module is included in the special IIMN build of MZmine. All available statistics were based on the spectral input file (mgf) and the resulting network file (graphml), which was downloaded from the dataset's corresponding GNPS results page. The graphml file contains all ion identity molecular networking results, namely, the nodes representing individual features and the edges between nodes. The mgf spectral summary file contains the corresponding $MS^2$ spectrum for each feature node. While classical MN and FBMN depend on $MS^2$ data for each node, IIN creates new $MS^1$-based edges that might include nodes without an $MS^2$ spectrum in the resulting network. For a comparison between FBMN and IIMN, only nodes present within both networks (with an $MS^2$ spectrum) are considered. A statistical summary and in-depth statistics on each dataset are provided in a supplementary Microsoft Excel workbook (Supplementary Data 1). Excerpts are summarized in Supplementary Table 2, and the different statistical measures and metadata items are described in Supplementary Table 5. One important measure is the identification density, i.e., all identified nodes and nodes with a maximum distance of n edges to at least one identified compound. Supplementary Figure 5 highlights how the additional edges of ion identity networking increase the identification density in the datasets, measured over a maximum distance of 1–5 edges. The increased density over one edge reflects the new links between unidentified to an identified node by IIN edge. The identification density is increased for 21 datasets, two datasets with poor identification rates exhibit no change, and one dataset lacks identifications. The maximum identification density increase of +8% over one edge results in a total of 42% of the nodes being either identified or directly linked to an identified compound. The network of the corresponding dataset, i.e., the post-column salt infusion study, contains a total of 22% identified nodes and 25% nodes with ion identity and $MS^2$ spectrum in 134 ion identity networks. Ion identity molecular networking decreased the number of unconnected singleton nodes by −12% to a total of 42%. Filtering out nodes with poor $MS^2$ spectra with less than four signals, which was used as the minimum number of signals for the library matching and FBMN networking, decreases the number of unconnected singleton nodes further to 29%. Consequently, the network contains many nodes without a match to any library or experimental spectra. Collapsing all nodes with IIN edges into molecular nodes reduces the total network size by −20%, which significantly reduces the overall redundancy and facilitates network visualization and analysis.

To extract the same statistics on any results from IIMN, download the networking results as a graphml file from a GNPS job page and use the mgf file of that analysis. The special MZmine IIMN build offers two modules in the Tools tab. More information and the latest IIMN enabled MZmine version are available (http://mzmine.github.io/iin_fbmn).

- GNPS results analysis (IIMN + FBMN)

   For a single analysis
   This tool also offers the extraction of new spectral library entries

- GNPS results analysis (IIMN + FBMN) of all sub

For multiple analyses at once
Generates statistics for each subfolder with exactly one graphml and mgf file (names do not have to match)

## IIMN-based spectral library generation

*From experimental datasets.* To comprehensively cover the fragmentation behavior of a molecule, spectral libraries should contain fragmentation spectra of different ion species acquired with different instrument types and fragmentation methods. IIMN might serve as a solution to expanded spectral libraries. To create new spectral library entries based on IIMN, all 24 datasets were searched for ion identity networks that contain a match to the GNPS spectral libraries with a minimum cosine similarity of 0.9 and a minimum number of shared fragment ions of 4–6, depending on each dataset's FBMN parameters. For each matching IIN, all contained ion identity features with an $MS^2$ spectrum and at least three signals above 0.1% relative intensity were extracted as new library spectra. The new library entries were constructed based on the highest library match and its attributes, namely, the compound name, structure strings as SMILES and InChI, and the neutral mass, the ion identity provided the ion species information and the precursor *m/z*, and dataset-specific metadata was added manually. With these strict rules, a total of 538 spectral entries were extracted from all 24 datasets. The new library has a broader and more distributed ion identity coverage when compared to selected representative spectral libraries from MassBank of North America (MoNA) and GNPS. At the same time, it is similar to spectral libraries that were generated with the new MSMS-Chooser library creation workflow in the GNPS ecosystem (Supplementary Fig. 5). The new IIMN-based library was made publicly available through the GNPS-library batch submission (Supplementary Tab. 3, https://gnps.ucsd.edu/ProteoSAFe/gnpslibrary.jsp?library=GNPS-IIMN-PROPOGATED).

*From a natural product compound library.* The library creation workflow was repeated and refined on the mass spectrometry data collected for the NIH NPAC ACONN collection of natural products (2179 compounds) provided by Ajit Jadhav (NIH, NCATS). The IIMN workflow was optimized and then applied to two LC-MS datasets collected on mass spectrometers operating in positive ionization mode, the MSV000080492 acquired on a qTOF-MS maXis II (Bruker Daltonics, GmbH) and the MSV000083472 acquired on a Q-Exactive (ThermoFisher Scientific, MA). During feature-based molecular networking, library matching was limited to the manually created GNPS libraries, which were based on the same qTOF-MS dataset (GNPS-NIH-NATURALPRODUCTSLIBRARY_ROUND2_POSITIVE, minimum matched signals = 3, minimum cosine similarity = 0.6). A new library for both datasets was created with new spectral entries with at least two signals above 0.1% relative intensity and with ion identities matching to the adduct of the library matches. Furthermore, library matches were filtered by a sample list of compound names contained in LC-MS samples. The IIMN library creation workflow resulted in 805 and 1314 new library entries for the qTOF-MS and the Q-Exactive datasets, respectively. The new IIMN-based library entries were made publicly available through the GNPS-library batch submission and merged into the existing manually created library GNPS-NIH-NATURALPRODUCTSLIBRARY_ROUND2_POSITIVE (Supplementary Table 3). In total, we generated 2,657 IIMN-based new spectral library entries.

*MZmine IIMN workflow for spectral library extraction.* To extract spectral library entries from any IIMN results, download the networking results as a graphml file from a GNPS job page and use the mgf file of that analysis. The special MZmine IIMN build offers the "GNPS results analysis" module in the Tools tab to create library entries based on these two files and provided metadata. The minimum GNPS-library match score sets a threshold for the extraction of library entries. Furthermore, library matches can be filtered to also match the ion identity to the adduct of the library match. A simple comparison between the different reporting formats for adducts was implemented. It removes all spaces, square brackets, and plus symbols (e.g., harmonizing M + H and $[M + H]^+$). Filters are available for new library entries with a minimum number of signals above a relative intensity threshold.

The latest information on the IIMN $MS^2$ library generation workflow in MZmine is available online:

http://mzmine.github.io/iin_fbmn
Documentation on the GNPS-library batch submission is available at:
https://ccms-ucsd.github.io/GNPSDocumentation/batchupload/

**Documentation**. The documentation of the IIMN workflow is shared in the GNPS documentations on GitHub (https://ccms-ucsd.github.io/GNPSDocumentation/fbmn-iin/), which also covers FBMN, classical MN, and other related tools. Suggested parameters for FBMN are described elsewhere[7] and the reproducible molecular networking protocol[5] describes MN parameters with step-by-step instructions. MZmine[17] provides help dialogs with parameter descriptions for each module and documentation links on their website (http://mzmine.github.io/

documentation.html). Tutorials and other references for MS-DIAL[18] are provided on their project website (http://prime.psc.riken.jp/compms/msdial/main.html). Bioconductor hosts the XCMS[16] and CAMERA[13] packages together with related information and their documentation (https://bioconductor.org/).

**Reporting summary**. Further information on research design is available in the Nature Research Reporting Summary linked to this article.

## Data availability

All raw (.raw) and centroided (.mzXML or .mzML) mass spectrometry data as well as processed data (.mgf and .csv) and ion identity molecular networks are available through the MassIVE repository (massive.ucsd.edu). Individual MassIVE dataset identifiers are listed in Supplementary Table 1. Dataset metadata and MZmine processing parameters are available in Supplementary Table 4. Links to IIMN job pages for each dataset are listed in Supplementary Table 6 with options for downloading or online analysis of results. Job cloning provides access to all parameter values and to reproducible data reanalysis. The statistical results on all 24 datasets are available in Supplementary Data 1. The IIMN-based $MS^2$ spectral libraries of propagated spectral entries can be used within GNPS or downloaded for free. The IIMN-based $MS^2$ spectral library from experimental datasets is available on GNPS (https://gnps.ucsd.edu/ProteoSAFe/gnpslibrary.jsp?library=GNPS-IIMN-PROPOGATED). The IIMN-based $MS^2$ spectral libraries from 2 datasets generated for the NIH Natural Products Library (NIH NPAC ACONN) were merged into the existing manually created GNPS library (NIH Natural Products Library Round 2), available on GNPS (https://gnps.ucsd.edu/ProteoSAFe/gnpslibrary.jsp?library=GNPS-NIH-NATURALPRODUCTSLIBRARY_ROUND2_POSITIVE). Individual download links for the two libraries are https://gnps.ucsd.edu/ProteoSAFe/status.jsp?task=c39d788d30f9408a9e53e68ec84868c6 (Q-Exactive dataset) and https://gnps.ucsd.edu/ProteoSAFe/status.jsp?task=904e6d42b5024c5cacef6dd86f02b714 (Q-TOF-MS dataset). The datasets are available with their accession IDs in the MassIVE repository: MSV000082081 [https://massive.ucsd.edu/ProteoSAFe/dataset.jsp?task=f65bfac6208a436fab483cd284f52a33], MSV000084116 [https://massive.ucsd.edu/ProteoSAFe/dataset.jsp?task=81538c459d0447ef972d267d2fb0111d], MSV000084008 [https://massive.ucsd.edu/ProteoSAFe/dataset.jsp?task=93f45e7eba2e456083a35a92610fff52], MSV000084099 [https://massive.ucsd.edu/ProteoSAFe/dataset.jsp?task=6ab3caf2593e4310a6516357f0657aeb], MSV00008 84119 [https://massive.ucsd.edu/ProteoSAFe/dataset.jsp?task=fb0ca514a168427d817a29ba90c4b9f2], MSV000084024 [https://massive.ucsd.edu/ProteoSAFe/dataset.jsp?task=ce840c3053d04c9b8ef1d6daf7068a98], MSV000082045 [https://massive.ucsd.edu/ProteoSAFe/dataset.jsp?task=64e3aacbbfd4b8681e7e788cb6b16fa], MSV000084101 [https://massive.ucsd.edu/ProteoSAFe/dataset.jsp?task=3b3d495b93c047adb2d8d25bf6205dc9], MSV000083729 [https://massive.ucsd.edu/ProteoSAFe/dataset.jsp?task=552dca562b1e4dd8a2ddbeb5e162d290], MSV000083772 [https://massive.ucsd.edu/ProteoSAFe/dataset.jsp?task=eaf6dfcfddc94612a4aa85e9c2e308db], MSV000083601 [https://massive.ucsd.edu/ProteoSAFe/dataset.jsp?task=48dc7250e98e45ffb988cfd353d53a3d], MSV0000840 45 [https://massive.ucsd.edu/ProteoSAFe/dataset.jsp?task=43e2ce9cd85c47678bf3250db b9e047b], MSV000084112 [https://massive.ucsd.edu/ProteoSAFe/dataset.jsp?task=a03639b5a08d42d283b714b64146087c], MSV000082630 [https://massive.ucsd.edu/ProteoSAFe/dataset.jsp?task=90cefc55f6464e20a873e471c5b962e1], MSV000084007 [https://massive.ucsd.edu/ProteoSAFe/dataset.jsp?task=c809f27dd91445f68c8cc522936119f4], MSV000084056 [https://massive.ucsd.edu/ProteoSAFe/dataset.jsp?task=bea6f8e8a5f14074bd884ee6dd659ab9], MSV000084107 [https://massive.ucsd.edu/ProteoSAFe/dataset.jsp?task=469779645cd94f159280a88e07c9cf7a], MSV000084063 [https://massive.ucsd.edu/ProteoSAFe/dataset.jsp?task=9d478e5f428443ff829f3decbd5759d6], MSV000084170 [https://massive.ucsd.edu/ProteoSAFe/dataset.jsp?task=3720644eead2496f8b48f6c09b8d4790], MSV000084134 [https://massive.ucsd.edu/ProteoSAFe/dataset.jsp?task=1ce9d2290ad04fc4bb4acf143ffe0b92], MSV000084148 [https://massive.ucsd.edu/ProteoSAFe/dataset.jsp?task=af4486f029a546d09e1dd572b66ddac5], MSV000084158 [https://massive.ucsd.edu/ProteoSAFe/dataset.jsp?task=1fedc205b6104024a901297a9c0ef151], MSV000084170 [https://massive.ucsd.edu/ProteoSAFe/dataset.jsp?task=17709af0ba294e1387192091a1c541e7], MSV000084118 [https://massive.ucsd.edu/ProteoSAFe/dataset.jsp?task=a13ac7b7be10421c8a168176752cc586]. Source data are provided with this paper.

## Code availability

The IIMN workflow is available as an interface on the GNPS web platform (https://gnps-quickstart.ucsd.edu/featurebasednetworking). The workflow code is open source and available on GitHub (https://github.com/CCMS-UCSD/GNPS_Workflows). It is released under the license of The Regents of the University of California and free for nonprofit research (https://github.com/CCMS-UCSD/GNPS_Workflows/blob/master/LICENSE). The workflow was written in Python (ver. 3.7) and deployed with the ProteoSAFE workflow manager employed by GNPS (http://proteomics.ucsd.edu/Software/ProteoSAFe/). We also provide documentation, support, example files, and additional information on the GNPS documentation website (https://ccms-ucsd.github.io/GNPSDocumentation/), and we invite everyone to contribute to the documentation on GitHub. The source code of all modules which were implemented into MZmine, e.g., the Export for IIMN module, the metaCorrelate grouping module, the ion identity networking modules, and the results and spectral library generation module, is available at http://mzmine.github.io/iin_fbmn under the GNU General Public License. The

source code for the custom GNPS export functions for XCMS is available at https://github.com/jorainer/xcms-gnps-tools under the GNU General Public License.

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

## Acknowledgements

We thank the German Chemical Industry Fund (FCI, Fonds der Chemischen Industrie) for a Ph.D. scholarship and travel support to R.S. P.C.D. was supported by the Gordon and Betty Moore Foundation (GBMF7622), the U.S. National Institutes of Health (P41 GM103484, R03 CA211211, R01 GM107550). We thank the Deutsche Forschungsgemeinschaft for support to D.P. (PE 2600/1-1), to H.U.H. (HU 730/9-3), and to S.B. and K.D. (BO 1910/20). L.F.N. was supported by the US National Institutes of Health (R01 GM107550), and the European Union's Horizon 2020 program (MSCA-GF, 704786). A.M.C.R. and P.C.D. were supported by the National Sciences Foundation grant IOS-1656481. L.I.A. was supported by the National Science Foundation grant OCE-1736656. T.P. was supported by the Czech Science Foundation Grant 21-11563 M. M.R. is supported by Public Health Service Grants AI126277, AI114625, AI145325, by the Chiba University-UCSD Center for Mucosal Immunology, Allergy, and Vaccines and an Investigator in the Pathogenesis of Infectious Disease Award from the Burroughs Wellcome Fund. D.P., M.A.P., and K.I.P. were supported by the National Science Foundation's Center for Aerosol Impacts on the Chemistry of the Environment (CAICE) under grant number CHE1801971. K.L.M. was supported by the Gordon and Betty Moore Foundation (GBMF6920) and the US National Institutes of Health (R01 GM132649). A.B. thanks FAPESP fellowship (2018/24865-4). C.O.G. thanks FAPESP scholarship (2019/06061-8). R.T. was supported by the US National Institutes of Health (NCCIH T32AT010131). A.A. O. acknowledges the support of Fulbright Commission and Consejo Nacional de Investigaciones Científicas y Técnicas (CONICET-Argentina). Z.K. was supported by the program Lumina Quaeruntur (LQ200202002) of the Czech Acad Sci. A.L.G. was supported by Vaincre la mucoviscidose and Association Grégory Lemarchal. H.M.R. thanks CNPq (#142014/2018-4), and the Brazilian Fulbright Commission for the scholarships provided. We thank Ajit Jadhav (NIH/NCATS) for providing the compounds used for the adduct induction experiment, and for the library generation. We thank Andreas J Andersson, Heather N Page, Travis A Courtney, Evan Fox, Sara P. Pucket, Kathleen E. Kyle, Jonathan L. Klassen and Marcy J. Balunas, Andrea Fidgett, and Michelle Gaffney for providing samples and assisting during sampling campaigns.

## Author contributions

General conceptualization: R.S., D.P., L.F.N., M.W., P.C.D. conceptualized the idea of IIMN and its integration into GNPS and feature-finding software tools. R.S., D.P., L.F.N., P.C.D. wrote the manuscript. R.S., B.A., F.H., H.U.H. conceptualized the MZmine feature grouping workflow. U.K., H.H. provided discussion and feedback on IIMN and the MZmine workflow. Development: R.S. developed the IIMN modules in MZmine and the MS² spectral library generation modules. M.W., R.S. developed the supplementary edges format in the FBMN workflow to enable IIMN. M.W. programmed the IIMN workflow on GNPS. R.S., M.W. developed the direct submission of MZmine data to run IIMN on GNPS. J.R., M.G.A. developed the XCMS/CAMERA IIMN integration in R. H.T. developed the MS-DIAL FBMN and IIMN integration. K.D. developed the MS² spectral merge function into the export modules for FBMN, IIMN, and SIRIUS, which was coordinated by S.B., T.P., A.K. provided feedback and help for the development and integration of IIMN in MZmine. Experiments, data analysis, and validation: D.P., L.F.N., A.A., A.A.O., G.A., A.B., A.T.A., A.M.C.R., J.M.G., E.C.G., C.O.G., Y.H., A.N.J., A.K.J., S. K., Z.K., I.K., A.L.G., K.L.M., M.N.E., M.A.P., M.W.P., R.T., F.V., K.W. performed experiments, analyzed data with the MZmine IIMN workflow, made data publicly available through MassIVE, and validated the results. K.A.P., M.R., H.Z., H.U.H., P.C.D. provided data and resources. R.S., D.P., A.T.A., A.N.J. analyzed data. A.T.A., R.S., A.N.J. wrote supplemental use cases. Y.H., S.K., A.N.J., A.K., B.A., Z.K. tested and provided feedback on the MZmine workflow. Documentation and videos: L.F.N., H.M.R., A.B., D. P., M.W., A.T.A., R.S., M.N.E. created the IIMN and FBMN documentations. R.S. produced video tutorials on FBMN, IIMN, and MZmine. M.W., R.S. produced videos on

FBMN and the direct submission of MZmine results to GNPS. D.P., M.W. produced a video tutorial for feature finding with MZmine and FBMN in GNPS. All authors contributed to the final manuscript.

## Competing interests

M.W. is the founder of Ometa Labs LLC. A.A. is a consultant for Ometa Labs LLC. S.B. and K.D. are co-founders of Bright Giant GmbH. A.K. is an employee of Bruker Daltonics GmbH & Co. KG.. P.C.D. is on the advisory board for Sirenas and Cybele. The other authors declare no competing interests.

## Additional information

[1]Institute of Inorganic and Analytical Chemistry, University of Münster, Münster, Germany. [2]Skaggs School of Pharmacy and Pharmaceutical Sciences, University of California San Diego, La Jolla, San Diego, CA, USA. [3]Scripps Institution of Oceanography, University of California San Diego, La Jolla, CA, USA. [4]CMFI Cluster of Excellence, Interfaculty Institute of Microbiology and Medicine, University of Tübingen, Tübingen, Germany. [5]Institute of Food Chemistry, University of Münster, Münster, Germany. [6]RIKEN Center for Sustainable Resource Science, Yokohama, Kanagawa, Japan. [7]RIKEN Center for Integrative Medical Sciences, Yokohama, Kanagawa, Japan. [8]Department of Biotechnology and Life Science, Tokyo University of Agriculture and Technology, Koganei-shi, Tokyo, Japan. [9]Institute for Biomedicine, Eurac Research, Affiliated Institute of the University of Lübeck, Bolzano, Italy. [10]Chair for Bioinformatics, Friedrich-Schiller-University, Jena, Germany. [11]Institute of Organic Chemistry and Biochemistry, Czech Academy of Sciences, Prague, Czech Republic. [12]Institute of Microbiology, Czech Academy of Sciences, Prague, Czech Republic. [13]Collaborative Mass Spectrometry Innovation Center, University of California San Diego, La Jolla, San Diego, CA, USA. [14]Institute of Biomedical Sciences, Universidade de São Paulo, São Paulo, SP, Brazil. [15]IRNASUS, Universidad Católica de Córdoba, CONICET, Facultad de Ciencias Agropecuarias, Córdoba, Argentina. [16]School of Pharmaceutical Sciences of Ribeirão Preto, Universidade de São Paulo, Ribeirão Preto, SP, Brazil. [17]Department of Pharmaceutical Sciences, College of Pharmacy, Oregon State University, Corvallis, OR, USA. [18]Univ. Grenoble Alpes, CNRS, Grenoble INP, CHU Grenoble Alpes, TIMC-IMAG, Grenoble, France. [19]Department of Psychiatry, University of California San Diego, San Diego, CA, USA. [20]NuBBE, Institute of Chemistry, , São Paulo State University (UNESP), Araraquara, SP, Brazil. [21]Division of Host-Microbe Systems & Therapeutics, Department of Pediatrics, University of California San Diego, La Jolla, CA, USA. [22]Chiba University-UC San Diego Center for Mucosal Immunology, Allergy and Vaccines (CU-UCSD cMAV), La Jolla, CA, USA. [23]These authors contributed equally: Robin Schmid, Daniel Petras, Louis-Félix Nothias. ✉email: pdorrestein@ucsd.edu

