## [Peer Review File · Nature Communications]

REVIEWER COMMENTS

Reviewer #1 (Remarks to the Author):

The manuscript describes the incorporation of MS1 molecular networking to account for the fact that during LC-MS data acquisition a given compound can result in multiple species arising from different ion adducts, multimer formation and in-source fragmentation. By coupling this MS1 molecular networking (which they term “ion identity molecular networking”, IIMN) with their established MS2 level molecular networking they can significantly reduce redundancy of LC-MS data and increase the % of molecular annotations and propagated annotations.

The approach is validated through the use of post-column solvent addition to induce specific adduct formation. It is further applied to publicly available datasets that nicely illustrate its utility. They provide extensive documentation to enable integration of this tool with the most commonly used feature finding platforms in metabolomics research. This will greatly facilitate use for many users.

The manuscript is very well written and easy to follow and understand. The functionality of the workflow they present will be of great interest to the field and will benefit many researchers. In particular they provide very interesting data illustrating the proportion of adduct species in different matrices, highlighting an underappreciated limitation of our current spectral libraries.

However, the approach they are implementing is not novel. They are basing the approach on ideas and workflows previously described and implemented - which they do cite appropriately. They do not claim to make significant improvements on these methods but instead just “make them work” in the context of their molecular networking platform. This does not diminish the importance of the tool, but it does question the appropriateness for publication in Nature Communications. It appears to be more of an “advertisement” of a new functionality of their tool rather than presentation of a novel idea.

Jessica Prenni

Reviewer #2 (Remarks to the Author):

In this manuscript the authors propose to use both LC-MS1 and MS2 chromatography data to improve chromatographic data annotation.

I think that the work shows a monumental effort in data acquisition, however, this is fundamentally a methodological paper and as such I feel that the manuscript falls short of technically describing the method itself. In that regard I am unable to recommend this manuscript for publication in Nature Communications because it is impossible for me to assess from the description that the manuscript and supplementary material provide the strengths and weaknesses of the proposed approach. Additionally, from the manuscript description I could not understand what the real advantage of using this approach is and, importantly, where it falls in the usual 'metabolomics workflow' and who would be the main user of this approach. Indiscriminate MS2 data is costly (in terms of time and money) and system dependent so it is unclear to me what extra advantage the approach provides besides the obvious expansion of the libraries of fragmentation spectra.

I will comment on different aspects that I found could be described in much more detail in the manuscript. If the authors are allowed and wish to resubmit the manuscript, my concerns might depend on the answers and methodological details.

The authors state in their abstract:

(..) These new relationships improve network connectivity, are shown to reveal novel ion ligand complexes, enhance annotation within molecular networks, and facilitate the expansion of spectral reference libraries.

My feeling is that several of the points they make in the abstract are not well addressed/described/shown in the manuscript.

improve network connectivity - In what sense do the authors think it is good to improve network connectivity? It is unclear what they mean by this, and it is also unclear to me that it is always advantageous to do so. It is clear that adding fragmentation spectra similarity data will increase the number of edges in the network. However, as the authors point out, adducts with the same delta m with respect to the neutral mass (i.e. sodium adduct) tend to be similar (because they have a sodium fragment that other adducts lack). Clearly, if the goal is to increase connectivity among adducts of the same precursor ion, that is not a desirable trait. This observation makes me wonder up to which

point adding indiscriminately fragmentation spectra similarity adds more noise than relevant information. Can novel ion-ligand complexes significantly distort the fragmentation pattern (with respect to say the H⁺ usual adduct for MS₂ in positive ionization)? If they do, are they still going to be able to detect similarities in the pattern of fragmentation? Adding on, the authors use cosine similarity to compare spectra. Are they using some threshold to decide whether two adducts are similar or not in terms of MS₂ spectra? If they are how are they selecting the threshold? If not how are the authors using that information?

enhance annotation within molecular networks - the authors do not provide any details about how they validate their choices (similarity in fragmentation + LC-MS₁ feature similarity) to properly identify the neutral mass of the ion and the molecular formula associated to a group of ions. I could not grasp from the text how the two types of information are combined and how the 'groups' are identified within the networks of molecular similarity. All of these choices are not arbitrary and should be selected using some sort of criterion, or some form of cross-validation analysis, which I have not found neither in the manuscript nor in the supplementary material.

In fact, it looks like (again it is unclear from the text) the authors are using CAMERA to do the neutral mass identification part. Other methods (like Clique-MS – ref 23) have been proposed to annotate LC-MS₁ data with superior performance, why are the authors using CAMERA? How much do their results depend on the precursor ion identification algorithm? Again it is unclear to me what exactly is the input data to CAMERA and what specific features of CAMERA the authors are using.

Also they authors do not show the improvement of using their approach with respect to not using it.

I think that this is a fundamental detail needed to show that the proposed approach can improve annotation.

novel ion ligand complexes – I agree that this is a feature that can be uncovered using this approach because MS₂ comparisons are able to discriminate this fact according to the authors. I think that this is the portion that is better reported in the manuscript and in the figures. However, I also feel that there is much that the authors do not explain. For instance, how well can the authors detect specific adducts? Can they clearly identify H⁺ adducts, or can they only detect specific ions which generate fragments that are typically not present in metabolites? Also, can the authors detect removal/addition of fragments such as water? The authors show that they can recover with their software the 'adduct' representation 'on average' at an adduct distribution level, but I would like to see what is the success at recovering each one of the adducts. Also, how can the authors obtain molecular formulas for fragments? Typically, there are many molecular formulas consistent with a specific neutral mass so again it is not so obvious to me that this step is straightforward. Also it is unclear to me if the authors are able to do identify novel adduct types in a targeted or untargeted way. Have they found these novel complexes after identifying the neutral mass/precursor ion or is it embedded in the process?

As a summary, I think that the authors have made a lot of work. However, it is not possible from the manuscript description the advantage that this tool represents and its limitations. I think that the authors should be clear about this by providing as many details as necessary to fully understand each one of the choices they are making.

Reviewer #3 (Remarks to the Author):

This paper tackles one of the key problems in the use of untargeted metabolomics: grouping the metabolic peaks from the same compound and merging them in the molecular network for metabolite identification.

Few papers in the past have attempted to solve this problem with different methods. The authors developed Ion Identity Molecular Networking that integrates chromatographic peak shape correlation analysis in the molecular networks to relate ion species of the same metabolites. This paper is unique and interesting by integrating MS1 information and MS2 information and then annotate them in the GNPS tool.

I have the following specific comments:

1. Using the IIMN, one feature may match different 'annotations' (namely ion identity). For example, feature 1 may be metabolite A with $[M+H]^+$ or metabolite B with $[M+Na]^+$. So how the IIMN solve this problem?
2. In Figure 1 d and Figure 2 C, the author said that the users can collapse ion identity networks into single molecular nodes to reduce complexity and redundancy. However, I can not find a detailed description of this step. If multiple features in one IIN connect with one feature (MS2 similarity), so after collapsing the IIN, which connection is used in the molecular network?
3. The authors used the 24 public datasets to validate the IIMN workflow, however, I don't find any analysis about the annotation correct rate of IIMN. So the question is how to know if the annotation results from MMRN are correct or not?

4. For the MN, we know it is based on MS2 similarity, but we know not all the features have MS2 spectra, so did those features without MS2 spectra be used in IIMN construction? If not, it would lose information for ion identify annotation. If yes, how connect those features with other features in MN?

5. Not a big deal, but I did not find a detailed description of IIMN workflow in the main text and method section. The “Ion identity molecular networking – workflow overview” in the method section is more like a tutorial of IIMN software. So a detailed description of the IIMN workflow may make the reader easier to understand the concept of the IIMN.

Response for NCOMMS-20-22211-T

First of all, we would like to thank the editor and the reviewers for their time and their constructive feedback. We appreciate their suggestions very much and addressed them in a revised manuscript and in the reply below. Our answers are indicated in blue and changes made in the manuscript are highlighted here in italic and referenced with the new line numbers in the revised manuscript are highlighted in bold.

Reviewer #1 (Remarks to the Author):

The manuscript describes the incorporation of MS1 molecular networking to account for the fact that during LC-MS data acquisition a given compound can result in multiple species arising from different ion adducts, multimer formation and in-source fragmentation. By coupling this MS1 molecular networking (which they term “ion identity molecular networking”, IIMN) with their established MS2 level molecular networking they can significantly reduce redundancy of LC-MS data and increase the % of molecular annotations and propagated annotations.

The approach is validated through the use of post-column solvent addition to induce specific adduct formation. It is further applied to publicly available datasets that nicely illustrate it’s utility. They provide extensive documentation to enable integration of this tool with the most commonly used feature finding platforms in metabolomics research. This will greatly facilitate use for many users.

The manuscript is very well written and easy to follow and understand. The functionality of the workflow they present will be of great interest to the field and will benefit many researchers. In particular they provide very interesting data illustrating the proportion of adduct species in different matrices, highlighting an underappreciated limitation of our current spectral libraries.

Thanks Jessica, for both your willingness to ID yourself and for your comments. Also, thanks a lot for your time reviewing the paper. We truly appreciate your acknowledgment of the validation and the importance of the documentation, as this is a critical part of making (software) tools broadly accessible to the community.

However, the approach they are implementing is not novel. They are basing the approach on ideas and workflows previously described and implemented - which they do cite appropriately. They do not claim to make significant improvements on these methods but instead just “make them work” in the context of their molecular networking platform. This does not diminish the importance of the tool, but it does question the appropriateness for publication in Nature Communications. It appears to be more of an “advertisement” of a new functionality of their tool rather than presentation of a novel idea.

Jessica Prenni

These comments are much appreciated, especially regarding the implementation and importance of the tool. It is indeed a new tool in the toolbox for users of the GNPS analysis ecosystem (used and accessed by thousands of users each month via >200,000 accessions per month, just Oct 2020 saw >332,000 accessions), it is also the first infrastructure that combines the well-established approaches of molecular networking with ion pattern discovery that can be leveraged in many additional and unique ways. Combining rule-based prioritization of co-eluting features by defined delta-masses, especially the combination with MS²-based molecular networking, represents an essential improvement of Molecular Networking as it overcomes issues arising from different CID fragmentation behavior of various ion adducts of the same molecule. We developed IIMN within GNPS as this is one of the aspects of molecular networking that is often asked about, especially as natural product scientists want to dereplicate as efficiently as possible and allow them to focus on new molecules. This is also why we added the IIMN obtained reference library of different adducts so people can more efficiently dereplicate and decrease the chances they go after known molecules even without performing IIMN. We also agree we could expand more on previous work here (within the limits of the journal specifications). To improve clarity, we expanded statements that highlight previous work regarding feature-correlation analysis and ion annotation in more detail. We highlight the utility of the current workflow, which this paper introduces, and highlight the significance of this improvement in the abstract and conclusion section.

New paragraphs in the introduction (**lines 110-132**) were also created based on the comments by reviewer two:

As various commonly detected ion adducts exhibit different fragmentation behavior during collisional activation (e.g., in collision-induced dissociation (CID) mode) (Supplementary Figure 1), MS² spectral networking on its own does not necessarily connect all ion adducts produced by a single compound. This often contributes to the unwanted separation of molecular families (sub-networks) and limits the propagation of library annotations through the networks. Especially the two ion species that are most frequently represented in spectral libraries ([M+H]⁺ and [M+Na]⁺, Supplementary Figure 8) typically stay unconnected. To overcome this central bottleneck of MN and to address the general problem of feature redundancy in MS-based metabolomics,^{1,2} we developed Ion Identity Molecular Networking (IIMN). IIMN fuses MS²-based spectral networks with an additional networking layer based on MS¹ feature shape correlation of identified ion species that originate from the same molecule.

Feature shape analysis as implemented in IIMN builds up on multiple tools that have been developed to identify ion species in LC-MS data. The first step, feature grouping, typically involves a retention time filter and the correlation of feature intensities across samples.³⁻⁵ Other tools, such as CAMERA and CliqueMS, add a pairwise correlation of feature shapes to the grouping metric.^{6,7} RAMClust provides an option to simultaneously process LC-MS data with MS² from data-independent acquisition (DIA).³ While many tools^{3,5-8} directly interoperate with the feature finding software XCMS,⁹ MS-FLO was developed to process exported feature lists from MZmine,¹⁰ MS-DIAL,¹¹ and XCMS. Generally, after feature grouping, ion species can be identified based on known mass differences. Connecting all ions that originate from the same molecule results in MS¹-based ion identity networks (IIN).

New paragraphs in the conclusion section (lines 223-244):

In conclusion, by establishing relationships between different ion species originating from the same compound and structurally similar compounds, IIMN facilitates molecular network interpretation and compound annotation. The combined networks with two layers of feature connectivity enable novel strategies to present and analyze mass spectrometry data visually. Especially networks with collapsed IIN reduce the redundancy of detecting multiple ion species per compound. IIMN successfully connected more related compound annotations in datasets from different analytical platforms and over a variety of small molecule compound classes. Hence, reducing the number of unconnected nodes and increasing the annotation density in molecular networks. An exciting application of IIMN is the expansion of spectral libraries by (re)-processing public datasets and propagating spectral library annotations to create library entries of connected ion identities. The identification of ion adducts can reveal novel ionophores, some of which will be biologically relevant and are still underappreciated in the function of small molecules^{12,13}. The integration into FBMN and the GNPS environment provided a platform to utilize IIMN in other related bioinformatics tools, e.g., SIRIUS¹⁴, CANOPUS¹⁵, and Qemistree¹⁶ for molecular formula and compound class level annotation, which will also facilitate additional validation of network connectivity. Furthermore, the open source code and generic connection between feature finding, ion identity molecular networking, and the online GNPS workflow encourage the implementation of interfaces to other feature grouping and ion identification algorithms. We anticipate that the new option to add orthogonal relationships between features to IIMN will stimulate the integration and development of additional tools for spectral alignment and measures of feature-feature relationships¹⁷.

Reviewer #2 (Remarks to the Author):

In this manuscript the authors propose to use both LC-MS1 and MS2 chromatography data to improve chromatographic data annotation.

I think that the work shows a monumental effort in data acquisition, however, this is fundamentally a methodological paper and as such I feel that the manuscript falls short of technically describing the method itself. In that regard I am unable to recommend this manuscript for publication in Nature Communications because it is impossible for me to assess from the description that the manuscript and supplementary material provide the strengths and weaknesses of the proposed approach. Additionally, from the manuscript description I could not understand what the real advantage of using this approach is and, importantly, where it falls in the usual 'metabolomics workflow' and who would be the main user of this approach. Indiscriminate MS2 data is costly (in terms of time and money) and system dependent so it is unclear to me what extra advantage the approach provides besides the obvious expansion of the libraries of fragmentation spectra.

Thanks to the reviewer for their time to provide detailed comments on the manuscript. We appreciate their comment regarding the technical description and pointing out the strengths and weaknesses of the manuscript. We agree with the reviewer that the paper lacked clarity and that we needed to improve it. We have now addressed the shortcomings that this reviewer highlighted,

especially to provide clarity of the technological side. The reviewer's input also motivated us to expand the ease-of-use and online documentation for this approach. Ultimately the problem we are solving here is one that everyone who analyzes MS²-based untargeted mass spectrometry data faces. First that there are multiple "analogs" of molecules commonly found in samples and that the spectral alignments do not differentiate between MS/MS of different ion adducts for a single molecule vs truly different but related molecules. In a molecular network, different ions for the same molecule often, but not always, reveal themselves as different molecular families. Second, we want to enable the annotation and collapsing of all ions forms of the same molecules so that they do not complicate downstream analysis.

The solution is via the merging of networks created of co-eluting LC-MS features and MS² spectral networks. As many in the community are not computer scientists, we implemented a workflow where non-coding labs can also access it via the web. The IIMN workflow is provided as an open source web tool and the computational resources can be accessed for free. We have addressed the points brought up by the reviewer in the point-to-point reply below and would love additional feedback if there are remaining aspects to the paper that need further improvement.

To expand on the technical description and documentation (also requested by the other reviewers as well as users) and to make it more clear to the reader how IIMN works and what the key benefits are in comparison to other MS and MS²-based metabolomics approaches, we expanded the abstract as well as the introduction and experimental concept section as follows:

Revised abstract (**lines 82-94**):

Molecular networking connects mass spectra of molecules based on the similarity of their fragmentation patterns. However, during ionization, molecules commonly form multiple ion species with different fragmentation behavior. As a result, the fragmentation spectra of these ion species often remain unconnected in tandem mass spectrometry-based molecular networks, leading to redundant and disconnected sub-networks of the same compound classes. To overcome this bottleneck, we developed Ion Identity Molecular Networking (IIMN) that integrates chromatographic peak shape correlation analysis into molecular networks to connect and collapse different ion species of the same molecule. The new feature relationships improve network connectivity for structurally related molecules, can be used to reveal novel ion-ligand complexes, enhance annotation within molecular networks, and facilitate the expansion of spectral reference libraries. IIMN has been integrated into various open source feature finding tools and the GNPS environment. Moreover, IIMN-based spectral libraries with a broad coverage of ion species were made publicly available.

New paragraph in the introduction (**lines 133-144**):

The IIMN workflow annotates and connects related ion species in feature-based molecular networks within the GNPS web platform. We integrate IIN into MS²-based molecular networks and demonstrate the application to LC-MS² studies that make use of product ion scans acquired in data-dependent acquisition (DDA) mode. The IIMN workflow comprises feature grouping, feature shape correlation, and identification of ion species using a variety of feature finding software tools such as MZmine,¹⁰ XCMS,⁹ and MS-DIAL¹¹ that make use of different algorithms for the identification of ion

adducts. A table of extracted MS^1 features, each with a consensus MS^2 spectrum, together with IIN results are then uploaded to GNPS to run the IIMN workflow on the web server. The resulting ion identity molecular networks contain two layers of feature (node) connectivity, linking ion identities of the same compound by MS^1 characteristics and structurally similar compounds by MS^2 spectral similarity (Figure 1).

New paragraph in the conclusion (**lines 223-231**):

In conclusion, by establishing relationships between different ion species originating from the same compound and structurally similar compounds, IIMN facilitates molecular network interpretation and compound annotation. The combined networks with two layers of feature connectivity enable novel strategies to present and analyze mass spectrometry data visually. Especially networks with collapsed IIN reduce the redundancy of detecting multiple ion species per compound. IIMN successfully connected more related compound annotations in datasets from different analytical platforms and over a variety of small molecule compound classes. Hence, reducing the number of unconnected nodes and increasing the annotation density in molecular networks.

New paragraph in the methods section (**lines 316-329**):

In general, the ion identity molecular networking (IIMN) workflow starts with LC- MS^2 data processing in one of the supported open source feature finding tools. After the creation of an aligned feature list of all samples, ion species that originate from the same analyte are grouped and annotated by MS^1 criteria, such as their retention time, feature shape correlation, and m/z difference. Here, such groups are named ion identity networks. Subsequently, information of detected features and their representative MS^2 spectra, ion identities, and connections to other ion identities are exported and transferred to the GNPS web server for the molecular networking part (refer to tool-specific sections for details). After the construction of ion identity molecular networks, features share connectivity based on MS^2 spectral cosine similarity and MS^1 -based feature shape correlation. In addition to this combined network, GNPS calculates a version with collapsed IIN, where one node represents multiple ions of the same molecule. Results are available in the GNPS web interfaces and as downloads in various open formats as tables and networking files to allow local visualization, reviewing, and post-processing.

New paragraph on the availability of documentation, which itself is also expanded (**lines 623-633**):

The documentation of the IIMN workflow is shared in the GNPS documentations on GitHub (<https://ccms-ucsd.github.io/GNPSDocumentation/fbmn-iin/>), which also covers FBMN, classical MN, and other related tools. Suggested parameters for FBMN are described elsewhere¹⁸ and the “reproducible molecular networking” protocol¹⁹ describes MN parameters with step-by-step instructions. MZmine¹⁰ provides help dialogs with parameter descriptions for each module and documentation links on their website (<http://mzmine.github.io/documentation.html>). Tutorials and other references for MS-DIAL¹¹ are provided on their project website (<http://prime.psc.riken.jp/compms/msdial/main.html>). Bioconductor hosts the XCMS⁹ and CAMERA⁶ packages together with related information and their documentation (<https://bioconductor.org/>).

In regards to the comment on the time and cost of indiscriminate MS² data, we are not completely sure if we understood the reviewer correctly. To the best of our knowledge, “indiscriminate MS² data” is a term not widely used in metabolomics. When it is used, it has been for data-independent acquisition (DIA) modes. IIMN however uses instead MS² product ion scans from data-dependent precursor ion selection. While IIMN theoretically can be adapted to include DIA, after spectrum deconvolution and parent mass prediction pre-processing (needed for molecular networking), the main target is for data from data-dependent MS² acquisition methods where the data has a precursor mass, an MS² spectrum, and a retention time feature. To make this more clear to a reader, we added an additional statement about the suggested and supported data types in the introduction (**lines 133-136**) as follows:

The IIMN workflow annotates and connects related ion species in feature-based molecular networks within the GNPS web platform. We integrate IIN into MS²-based molecular networks and demonstrate the application to LC-MS² studies that make use of product ion scans acquired in data-dependent acquisition (DDA) mode.

I will comment on different aspects that I found could be described in much more detail in the manuscript. If the authors are allowed and wish to resubmit the manuscript, my concerns might depend on the answers and methodological details.

Thanks for pointing to aspects that were not clear - this is really appreciated and helpful.

The authors state in their abstract:

(..) These new relationships improve network connectivity, are shown to reveal novel ion ligand complexes, enhance annotation within molecular networks, and facilitate the expansion of spectral reference libraries.

My feeling is that several of the points they make in the abstract are not well addressed/described/shown in the manuscript.

improve network connectivity - In what sense do the authors think it is good to improve network connectivity? It is unclear what they mean by this, and it is also unclear to me that it is always advantageous to do so.

Thanks a lot for pointing this out. We have aimed to clarify within the constraint of journal length. In general, we aim for and typically experience that molecular networking successfully connects features (nodes) from similar compound classes based on the cosine similarity of MS² fragmentation spectra. However, this is most often only the case for features of similar ion adducts (e.g., [M+H]⁺, [M+NH₄]⁺, and [M-H₂O+H]⁺). Other ion species often produce fragmentation patterns that remain unconnected by MS² spectral similarity (e.g., [M+Na]⁺ to [M+H]⁺). With IIMN we are linking these independent networks allowing the user to improve the insights obtained through improved understanding of propagated annotations or even improved understanding of the global molecular diversity of the samples.

To make this more clear in the manuscript, we expanded and rewrote the following paragraph in the introduction (**lines 110-121**) as follows:

As various commonly detected ion adducts exhibit different fragmentation behavior during collisional activation (e.g., in collision-induced dissociation (CID) mode) (Supplementary Figure 1), MS² spectral networking on its own does not necessarily connect all ion adducts produced by a single compound. This often contributes to the unwanted separation of molecular families (sub-networks) and limits the propagation of library annotations through the networks. Especially the two ion species that are most frequently represented in spectral libraries ([M+H]⁺ and [M+Na]⁺, Supplementary Figure 8) typically stay unconnected. To overcome this central bottleneck of MN and to address the general problem of feature redundancy in MS-based metabolomics,^{1,2} we developed Ion Identity Molecular Networking (IIMN). IIMN fuses MS²-based spectral networks with an additional networking layer based on MS¹ feature shape correlation of identified ion species that originate from the same molecule.

Since the original submission, we have expanded the new method in the GNPS web-platform to directly collapse ion identity networks (IIN) in the final IIMN networking output thereby reducing the complexity of interpretation of the data collected by untargeted mass spectrometry. We expanded on the technical aspects of IIN collapsing in a new paragraph in the methods section (**lines 360-383**) and updated Supplementary Figure 5 to highlight new visualization options:

1. [...]
9. The option "Download Cytoscape Data" provides two .graphml networking files
 - a. The standard FBMN and IIMN networks (base directory)
 - b. IIMN networks with collapsed ion identity networks (in "gnps_molecular_network_iin_collapse_graphml" directory)
10. The option "Direct Cytoscape Preview/Download" provides the IIMN network and its collapsed version as Cytoscape projects with various style presets

Generation of collapsed ion identity networks

One result of the GNPS IIMN workflow is the combined networks with IIN collapsed into single nodes. For this, all ion nodes with the same IIN ID are merged into a representative node based on the feature with the highest library match score, if available, or feature abundance. While all IIN edges are collapsed, MN edges of all ion identities are redirected to their representative nodes so that duplicates replace existing edges if their edge score (cosine similarity) is higher. Limiting the number of MN edges to the one with the highest cosine similarity. Furthermore, representative collapsed nodes are extended by multiple attributes, including the intensity of each ion identity and their summed intensity. This enables the direct comparison of ionization tendencies and provides new visualization options. An example with pie-charts of the ion abundances is demonstrated in Supplementary Figure 5.

Supplementary Figure 5: Ion identity molecular networking results for *Stachybotrys chartarum* liquid culture extracts (MSV000084134). The presented two visualization options compare the combined ion identity molecular network (a), with all ion identities of the same compound sorted in circles, and a collapsed IIN network version (b), with pie-charts representing the relative ion abundances of the corresponding compounds. Compounds A and D were annotated by spectral library match and compounds A, B, D, and E were verified by nuclear magnetic resonance (NMR) spectroscopy²⁰. Modifications between compounds are based on the compound structures or the differences of the average neutral masses for each IIN. The modifications of +O and +H₂O from B to C to F can also be deduced from the ion identity annotations, as the maximum in-source water losses for each compound confirm the addition of oxygens; Compounds A, B, and D (-2H₂O maximum water losses), compound C (-3H₂O), and compound F (-4H₂O). Therefore, this example proves that IIN can yield structure relevant information which facilitates structure annotation and propagation.

It is clear that adding fragmentation spectra similarity data will increase the number of edges in the network. However, as the authors point out, adducts with the same delta m with respect to the neutral mass (i.e. sodium adduct) tend to be similar (because they have a sodium fragment that other adducts lack). Clearly, if the goal is to increase connectivity among adducts of the same precursor ion, that is not a desirable trait. This observation makes me wonder up to which point adding indiscriminately fragmentation spectra similarity adds more noise than relevant information.

Thanks for the comment. One should reverse the thinking. Molecular networking, a tool that is used by thousands of investigators worldwide, is commonly used to discover analogs of molecules and for the dereplication of followup compound identification experiments. One of the common nuances and real frustrations by members of the community that perform molecular networking is that different ion species may form different molecular families in the resulting networks; This is unavoidable. By integrating molecular networks with ion identity networks, it is possible to increase the connectivity of identical and related structures. This, however, has not been possible before. For example, an in-source fragment ion may connect to a non-in-source fragment of another structurally similar molecule through molecular networking but neither have annotations. Then propagation of annotations can often be achieved via the IIN edges instead.

We have discussed the “indiscriminate data” issue above in a previous comment but in summary - Ion identity networking is based on MS¹ data and molecular networking on MS² spectra which are typically acquired with DDA methods. Processing of indiscriminate data-independent acquisition (DIA) MS² spectra is out of scope of this manuscript.

To emphasize the default data acquisition mode more, we added a statement in the introduction (**lines 133-136**) as follows:

The IIMN workflow annotates and connects related ion species in feature-based molecular networks within the GNPS web platform. We integrate IIN into MS²-based molecular networks and demonstrate the application to LC-MS² studies that make use of product ion scans acquired in data-dependent acquisition (DDA) mode.

Can novel ion-ligand complexes significantly distort the fragmentation pattern (with respect to say the H⁺ usual adduct for MS² in positive ionization)? If they do, are they still going to be able to detect similarities in the pattern of fragmentation?

Yes and this is the most important point of the paper. With this question, the reviewer correctly captures the key reason for the development of IIMN. While “Soft” ions of the same and similar molecules ([M+H]⁺, [M+NH₄]⁺, [2M+H]⁺, [M-H₂O+H]⁺, ...) often result in similar fragmentation spectra and connections based on MN. Other ion adducts ([M+Na]⁺, [M+K]⁺, [M+Ca-H]⁺, [M+Fe-2H]⁺, [M+Zn-H]⁺) are more likely to lead to dissimilar fragmentation spectra that stay unconnected to MS² spectra of other ion adducts. Supplementary Figure 1 compares the fragmentation spectra of a variety of molecules and their M+H and M+Na adducts. Multiple figures (Figure 2 (a-c) and Supplementary Figures 4 and 5) showcase the separation of ion adducts into individual MS² molecular networks (blue edges - MS² cosine similarity).

Since this was not immediately clear to the reviewer we have aimed to clarify this key part by rewriting the following paragraph in the abstract and in the introduction as follows:

Abstract (lines 82-89):

Molecular networking connects mass spectra of molecules based on the similarity of their fragmentation patterns. However, during ionization, molecules commonly form multiple ion species with different fragmentation behavior. As a result, the fragmentation spectra of these ion species often remain unconnected in tandem mass spectrometry-based molecular networks, leading to redundant and disconnected sub-networks of the same compound classes. To overcome this bottleneck, we developed Ion Identity Molecular Networking (IIMN) that integrates chromatographic peak shape correlation analysis into molecular networks to connect and collapse different ion species of the same molecule.

Introduction (lines 110-121):

As various commonly detected ion adducts exhibit different fragmentation behavior during collisional activation (e.g., in collision-induced dissociation (CID) mode) (Supplementary Figure 1), MS² spectral networking on its own does not necessarily connect all ion adducts produced by a single compound. This often contributes to the unwanted separation of molecular families (sub-networks) and limits the propagation of library annotations through the networks. Especially the two ion species that are most frequently represented in spectral libraries ([M+H]⁺ and [M+Na]⁺, Supplementary Figure 8) typically stay unconnected. To overcome this central bottleneck of MN and to address the general problem of feature redundancy in MS-based metabolomics,^{1,2} we developed Ion Identity Molecular Networking (IIMN). IIMN fuses MS²-based spectral networks with an additional networking layer based on MS¹ feature shape correlation of identified ion species that originate from the same molecule.

We love to hear if this newly written description brings out this key point more clearly, if not then we have to provide further clarity.

Adding on, the authors use cosine similarity to compare spectra. Are they using some threshold to decide whether two adducts are similar or not in terms of MS² spectra? If they are how are they selecting the threshold? If not how are the authors using that information?

Thanks a lot for this question. In molecular networking, we typically use parameters for spectral library matching by cosine similarity that correspond to a false discovery rate (FDR) of 1%^{21,22} as a starting point. Co-authors used their dataset specific MN parameters for the different projects. We have now provided a table of the GNPS job links for each project. Cloning of each job provides access to all parameters and to reproducible data reanalysis. This was now clarified in the manuscript's data availability section **(lines 658-660):**

Links to IIMN job pages for each dataset are listed in Supplementary Table 6 with options for downloading or online analysis of results. Job cloning provides access to all parameter values and to reproducible data reanalysis.

And we added Supplementary Table 6 to the Supplemental Information (with the caption below):

Supplementary Table 6: *Links to the GNPS IIMN job pages for each dataset provide options for downloading the results and analyzing them in the GNPS web interfaces. The IIMN workflow parameters are available by job cloning, which further enables reanalysis.*

enhance annotation within molecular networks - the authors do not provide any details about how they validate their choices (similarity in fragmentation + LC-MS1 feature similarity) to properly identify the neutral mass of the ion and the molecular formula associated to a group of ions. I could not grasp from the text how the two types of information are combined and how the 'groups' are identified within the networks of molecular similarity.

Sorry that this was not clear in the manuscript. The compound annotations are derived from MS² spectral library matching during molecular networking. As part of the ion identity molecular networking workflow, we search more than 30 public spectral reference libraries, including all three MassBanks (MONA, EU, and JP), ReSpec, EMBL library, CASMI, GNPS contributed, HMDB etc. (<https://gnps.ucsd.edu/ProteoSAFe/libraries.jsp>) and NIST17.

We have changed the introduction (**lines 99-104**) to cite the MN paper²¹ and MN protocol¹⁹:

MN relies on the principle that similar structures tend to form similar patterns in fragmentation mass spectra (MS²). MN is built up through the pairwise spectral comparisons of a dataset, creating an MS² spectral network. This network is then enriched by annotating the experimental MS² spectra against MS² spectral libraries^{19,21} or compound databases (Figure 1). In the resulting molecular networks, annotations can be propagated through the network edges to adjacent nodes²³.

SIRIUS 4¹⁴ provided molecular formulas for the zinc-binding project based on MS¹ isotope pattern and MS² fragmentation spectra (Supplementary Figure 3). Nuclear magnetic resonance spectroscopy yielded structures for the *Stachybotrys chartarum* dataset (Supplementary Figure 5).

We have now made sure that we clarified that the molecular formulas in the zinc-binding project are created using SIRIUS 4 and added the following text in Supplementary Note 1 (**lines 68-70**):

Molecular formulas were predicted with SIRIUS 4.0 after exporting MS¹ isotope pattern and MS² spectra as a data processing step in MZmine.

IIMN combines annotations from MS1 ion identity networking and MS2 spectral library matching and provides downloadable networks and tables for manual and programmatic downstream analysis. A newly added function in the GNPS workflow adds a cross-validation step to the standard output by comparing the ion identity to the ion species field of a corresponding library match. We have added the following paragraph (also in regards to comment 3 of reviewer 3) to describe the new cross-validation of ion identities and library matches to the methods section in the „Ion identity molecular networking – workflow overview“ (**lines 385-406**).

Cross-validation of MS² spectral library matches and MS¹ ion identities

In IIMN, nodes may combine annotations from MS² spectral library matching and MS¹ ion identity networking. As cross-validation, GNPS parses and harmonizes the ion species string of both the detected ion identity and matching spectral library entry before checking for equality. The results are reported as an additional column in the node table. This equality check facilitates manual reviewing and the spotting of discrepancies between the MS¹ and MS² annotations.

The ion string parser harmonizes an input (e.g., [M-H₂O+2H]²⁺) in the following steps:

- 1. Spaces are removed*
- 2. Charge state is detected and removed from the input (2+)*
- 3. Brackets are removed ([]())*
- 4. Input is split into added (+2H) and removed (-H₂O) parts*
- 5. Both lists are sorted alphabetically (+2H sorted by letter H)*
- 6. If the charge state is missing, it is calculated for all parts that are listed in a lookup table (e.g., +Na or +H correspond to charge 1+)*
- 7. The harmonized string is constructed by concatenation of [M-all removed parts+all added parts]charge state.*

As an example, the harmonized string [M+H]⁺ is produced by the input strings M+H, M+H+, and [M+H]⁺, which are all commonly found in the GNPS spectral libraries and as an output of various software tools.

The full open source code of the ion string parser and its latest charge lookup table can be found on GitHub (https://github.com/CCMS-UCSD/GNPS_Workflows).

All of these choices are not arbitrary and should be selected using some sort of criterion, or some form of cross-validation analysis, which I have not found neither in the manuscript nor in the supplementary material.

For feature-based molecular networking the suggested parameters are described in the FBMN paper¹⁸ and a good starting point for molecular networking parameters can be found as step-by-step instructions in the “reproducible molecular networking” protocol¹⁹ and in the documentation (<https://ccms-ucsd.github.io/GNPSDocumentation/featurebasedmolecularnetworking/>). We routinely use cross-validation studies where appropriate and that requires a model of some sort. In this case, since there is no model that we build from the combination of MS¹ and MS² data there is no cross-validation possible. Instead, our validation of the enhanced networks is via the use of experimental data specifically acquired for this purpose. For example, with the addition of different salt solutions, we show that the corresponding ion adducts increase in intensity and frequency which is a key validation. In addition, we demonstrate that IIMN can be used to find new structures as validated with nuclear magnetic resonance spectroscopy yielded structures for the *Stachybotrys chartarum* dataset (Supplementary Note 3, Supplementary Figure 5). There are also two papers in review, where we used IIMN to solve a longstanding question as to how *E. coli* Nissle used zinc to protect against salmonella infections²⁴ and how a traditional Samoan medicine suppresses inflammation-through a molecular family of specific metal-small molecule ion pairs (under review).

We have added a documentation section in the methods to point readers to related resources that explain the workflows and their parameters (**lines 623-633**)

Documentation

The documentation of the IIMN workflow is shared in the GNPS documentations on GitHub (<https://ccms-ucsd.github.io/GNPSDocumentation/fbmn-iin/>), which also covers FBMN, classical MN, and other related tools. Suggested parameters for FBMN are described elsewhere¹⁸ and the “reproducible molecular networking” protocol¹⁹ describes MN parameters with step-by-step instructions. MZmine¹⁰ provides help dialogs with parameter descriptions for each module and documentation links on their website (<http://mzmine.github.io/documentation.html>). Tutorials and other references for MS-DIAL¹¹ are provided on their project website (<http://prime.psc.riken.jp/compms/msdial/main.html>). Bioconductor hosts the XCMS⁹ and CAMERA⁶ packages together with related information and their documentation (<https://bioconductor.org/>).

In fact, it looks like (again it is unclear from the text) the authors are using CAMERA to do the neutral mass identification part.

Thanks for bringing this up. IIMN works with different feature finding tools that make use of different correlation modules (Eg. MZmine makes use of metaCorrelate whereas XCMS is using CAMERA). This was detailed in the methods section. As this might have been a little hidden, we moved the info now in the main text which now reads as follows (**lines 171-179**):

To test the workflow with data generated from various sample types and on different experimental platforms, 24 public datasets were processed using the MZmine workflow and its metaCorrelate algorithm for feature grouping and ion identity networking (Figure 2g, Supplementary Table 1). All the specific parameters for processing are provided in the methods section (“Dataset processing”) and Supplementary Table 5 and links to the corresponding IIMN job pages on GNPS are listed in Supplementary Table 6. MZmine feature finding parameters were optimized for each dataset by various coauthors, while the feature grouping and ion identity networking parameters were kept constant for better comparability.

Other methods (like Clique-MS – ref 23) have been proposed to annotate LC-MS1 data with superior performance, why are the authors using CAMERA? How much do their results depend on the precursor ion identification algorithm? Again it is unclear to me what exactly is the input data to CAMERA and what specific features of CAMERA the authors are using.

We agree that there are many more algorithms for MS¹ annotation of ion species. Other open or proprietary LC-MS processing tools use various MS¹ annotation algorithms producing different results and output formats. Currently, IIMN supports the MZmine/metaCorrelate algorithms, XCMS/CAMERA, and MS-DIAL to reach a broad user base. Different co-authors helped to make the output of these tools compatible with the current GNPS workflow. We also outline the needed input to interface any MS¹ annotation tool to our online IIMN workflow on GNPS. CliqueMS is an

interesting comprehensive approach and we would love to help with the integration into our workflow. As CliqueMS interoperates with XCMS, the CAMERA support might serve as an example for reformatting the XCMS/CliqueMS output.

To better review the literature and capture tools for the annotation of ion species, we added a paragraph to the introduction (**lines 122-132**):

Feature shape analysis as implemented in IIMN builds up on multiple tools that have been developed to identify ion species in LC-MS data. The first step, feature grouping, typically involves a retention time filter and the correlation of feature intensities across samples.³⁻⁵ Other tools, such as CAMERA and CliqueMS, add a pairwise correlation of feature shapes to the grouping metric.^{6,7} RAMClust provides an option to simultaneously process LC-MS data with MS² from data-independent acquisition (DIA).³ While many tools^{3,5-8} directly interoperate with the feature finding software XCMS,⁹ MS-FLO was developed to process exported feature lists from MZmine,¹⁰ MS-DIAL,¹¹ and XCMS. Generally, after feature grouping, ion species can be identified based on known mass differences. Connecting all ions that originate from the same molecule results in MS¹-based ion identity networks (IIN).

Also they authors do not show the improvement of using their approach with respect to not using it. I think that this is a fundamental detail needed to show that the proposed approach can improve annotation.

The challenge is that the comparison is a manual interpretation of molecular networks to match parent masses that represent the same molecule-which is what the community does at this time when inspecting molecular networking jobs. It would take days/weeks of manual inspection vs doing it computationally. Manual interpretation of molecular networks is hard to quantify but we would welcome suggestions if the reviewer knows how to do this in an efficient and representative manner for diverse datasets? Because feature-based molecular networking was only recently developed, a key advancement for IIMN to be possible, it has only now begun to be possible to do this computational comparison now. What we show is that more data presented in the ion identity molecular network can be explained. Furthermore, IIMN can be used to reduce the complexity of the molecular networks by collapsing IINs into single nodes. Finally, the increased connectivity can provide more propagated annotations-or hypothetical relationships to unknown molecules, something that is very desirable by the natural product community, which would then need to be verified by other means such as NMR after targeted isolation as we showed in this paper for the *Stachybotrys chartarum* dataset (Supplementary Figure 5).

As a visual example for a complex experimental dataset, Supplementary Figure 4 directly compares the results for FBMN and IIMN. While FBMN connects many features of structurally related compounds by their MS² similarity, a significant number of annotated bile acids do not fall into the same sub-network. IIMN solves the issue of missing links by adding connections between ion adducts of the same molecule. Now, free and conjugated bile acids fall into a network with a higher annotation density and more connections between related compounds. Similar examples for other compound classes are given on a smaller scale in Figure 2 (a-c) and Supplementary Figure 5. The

annotation density was shown to increase from FBMN to IIMN (Supplementary Figure 7) with the number of unconnected nodes decreasing. The challenge to assess and quantify the reliability of connecting nodes from similar compounds remains for both molecular networking and ion identity networking. A way to address this general issue, would be to measure the structural similarity of compounds annotated by spectral library matching and tools for in-silico compound class annotation.

To point this out in the paper we added the following statement in the discussion section (**lines 235-238**):

The integration into FBMN and the GNPS environment provided a platform to utilize IIMN in other related bioinformatics tools, e.g., SIRIUS¹⁴, CANOPUS¹⁵, and Qemistree¹⁶ for molecular formula and compound class level annotation, which will also facilitate additional validation of network connectivity.

To emphasize the difference between results from FBMN and IIMN more, we have adjusted the discussion section in the following paragraph (**lines 184-195**):

*In a dataset with 88 animal bile acid extracts, multiple smaller networks and unconnected nodes were combined to a large network of free bile acids and those conjugated to amino acids or sulfate, resulting in higher network connectivity for IIMN over FBMN (Supplementary Note 2, Supplementary Figure 4). IIMN also yielded additional structural information in the case of mold samples from *Stachybotrys chartarum* (Supplementary Note 3, Supplementary Figure 5). In this project, IIMN revealed novel phenylspirodrimane derivatives which were verified by nuclear magnetic resonance spectroscopy (NMR).²⁰ In the network, the increasing number of aliphatic hydroxyl groups was reflected by the maximum number of in-source water losses, whereas acetylation of hydroxy groups reduced this number. The manual inspection of IIMN results was facilitated by additional MS¹ annotations provided by ion nodes that lack MS² fragmentation data and are consequently unavailable to the FBMN workflow.*

We further included a statement in the methods section to clarify that IIMN introduces the option to retain and connect features that lack MS² fragmentation data. This is a significant advancement compared to FBMN, as only a fraction of detected ions trigger MS² acquisition in data-dependent acquisition (DDA) experiments (**lines 341-345**).

In comparison to FBMN, IIMN can include features that are lacking MS² fragmentation spectra but are connected to other feature nodes by MS¹ IIN edges. Regarding a higher detectability by MS¹ compared to triggered MS² acquisition, the additional nodes with ion identities complement the resulting networks with information otherwise lost in FBMN or classical MN.

novel ion ligand complexes – I agree that this is a feature that can be uncovered using this approach because MS² comparisons are able to discriminate this fact according to the authors. I think that this is the portion that is better reported in the manuscript and in the figures.

Thanks-this is indeed one of the many possible use cases.

However, I also feel that there is much that the authors do not explain. For instance, how well can the authors detect specific adducts? Can they clearly identify H + adducts, or can they only detect specific ions which generate fragments that are typically not present in metabolites?

It is true that the identification of in-source ion adducts is not trivial. To address this, we enable IIMN with the option to start in one of three feature finding tools, each with different algorithms to identify ion adducts (MZmine/metaCorr, XCMS/CAMERA, and MS-DIAL). At this point, ions are mainly identified based on the MS¹ data with only limited information extracted from the MS² fragmentation spectra, e.g., to verify multimeric ion species. Therefore, we are only able to identify a group of ions with at least one descriptive *m/z* difference. The difference of -H₂O alone will not identify the complete adduct composition but rather point to an unknown adduct as [M-H₂O+?]⁺. For the identification of in-source water losses, it is important to have at least another different adduct (often [M+Na]⁺). The metaCorrelate algorithm in MZmine (which was used throughout the manuscript) only checks the MS² fragmentation data to verify multimeric ion species by their characteristic -M fragments (e.g., [2M+H]⁺→[M+H]⁺). Moreover, it is possible that one feature is annotated as different ion species with multiple co-eluting features.

We have expanded the “IIMN with MZmine” part in the methods section to better describe the handling of ion identities in MZmine (**lines 414-420**).

Both the creation and expansion of ion identity networks follow customizable lists of adducts and in-source modifications to cover any type of multimers, in-source fragments, and adducts. The IIN procedure lists all possible ion identity pairs between two features and ranks them according to the maximum number of features that support a specific annotation, i.e., the ion identity network size. While a feature might be annotated as two different ion species that point to different metabolites, the current workflow will only create additional IIN edges and ion species metadata for the highest-ranking ion identity per feature. This filter decreases the number of spurious matches.

Also, can the authors detect removal/addition of fragments such as water?

Yes, there are quite a few examples that we show in the paper where we link in-source water loss fragment ions (captured by MS¹ correlation) or analogs of molecules (captured via molecular networking) but we did not make this obvious. Networking connections of such adducts are shown in Figure 2a-b and Supplementary Figure 5, denoted as [M-H₂O+H]⁺ or [M-nH₂O+H]⁺ indicating one or more water losses. Figure 2g shows the relative number of in-source water loss ions compared to other commonly detected ions.

We adjusted parts of the discussion to underline the relevance that in-source water losses had on one of our project datasets (**lines 193-195**):

In this project, IIMN revealed novel phenylspirodrimane derivatives which were verified by nuclear magnetic resonance spectroscopy (NMR).²⁰ In the network, the increasing number of aliphatic hydroxyl groups was reflected by the maximum number of in-source water losses, whereas acetylation of hydroxy groups reduced this number. The manual inspection of IIMN results was

facilitated by additional MS¹ annotations provided by ion nodes that lack MS² fragmentation data and are consequently unavailable to the FBMN workflow.

The authors show that they can recover with their software the ‘adduct’ representation ‘on average’ at an adduct distribution level, but I would like to see what is the success at recovering each one of the adducts.

In Supplementary Figure 5b, ion identity networks were collapsed which reduces the complexity of the network and demonstrates more clearly the importance of water loss in-source fragmentation in different related molecules (at different retention times) by mapping molecular networking edges and MS¹ IIN edges more clearly in the figure for stachybotrys chartarum liquid culture extracts. Regarding the success of recovering adducts, we now show that upon infusion of different salt solutions, the change of detectability of the corresponding adducts is statistically robust with p-values < 0.001 for the [M+Na]⁺ ion identity, between the control and the infusion of NaAcetate, and for the [M+NH₄]⁺ and [M+ACN+NH₄]⁺ ion identities, between the control and the infusion of NH₄Acetate. Interestingly, the usually unexpected [M+ACN+NH₄]⁺ exclusively forms after the addition of NH₄Acetate, which further verifies ion identities connected in the same IIN.

We have changed the first paragraph of the discussion section with more details on the validation of ion identities by post-column salt infusion (**lines 162-170**):

Finally, the abundance change of identified adducts ([M+H]⁺, [M+NH₄]⁺, [M+Na]⁺) in our benchmark dataset is in agreement with the different post-chromatography salt infusion conditions (H₂O, Na-Acetate, or NH₄-Acetate, Figure 2f) which validates ion species identification on a dataset level. For instance, the abundance of [M+Na]⁺ and [M+NH₄]⁺ ion identities was significantly (p<0.001) higher in the corresponding samples with the post-column infusion of sodium acetate or ammonium acetate, respectively, when compared to the control samples. The exclusive formation of an uncommon [M+ACN+NH₄]⁺ in-source cluster after infusion of ammonium ions further verifies connected ion identities.

Supplementary Figure 5: Ion identity molecular networking results for *Stachybotrys chartarum* liquid culture extracts (MSV000084134). The presented two visualization options compare the combined ion identity molecular network (a), with all ion identities of the same compound sorted in circles, and a collapsed IIN network version (b), with pie-charts representing the relative ion abundances of the corresponding compounds. Compounds A and D were annotated by spectral library match and compounds A, B, D, and E were verified by nuclear magnetic resonance (NMR) spectroscopy²⁰. Modifications between compounds are based on the compound structures or the differences of the average neutral masses for each IIN. The modifications of +O and +H₂O from B to C to F can also be deduced from the ion identity annotations, as the maximum in-source water losses for each compound confirm the addition of oxygens; Compounds A, B, and D (-2H₂O maximum water losses), compound C (-3H₂O), and compound F (-4H₂O). Therefore, this example proves that IIN can yield structure relevant information which facilitates structure annotation and propagation.

Also, how can the authors obtain molecular formulas for fragments? Typically, there are many molecular formulas consistent with a specific neutral mass so again it is not so obvious to me that this step is straightforward.

Correct, because the molecular formula generation tools are described elsewhere we are not spending much time on this in this paper, especially since we are space limited. In the zinc-binding project, we obtained the molecular formulas using SIRIUS 4¹⁴ on the MS¹ isotope pattern and MS² spectra after processing in MZmine. SIRIUS works by assigning all possible molecular formulas to each fragment ion and then uses the best combination of all combinations of molecular formulas to predict the best molecular formula for the parent mass. We recently also introduced ZODIAC to complement SIRIUS 4 that uses Bayesian statistics to derive the molecular formula, which is now also compatible with the IIMN workflow.²⁵

We have now made sure that we clarified that the molecular formulas are created using SIRIUS 4 and added the following to the text in Supplementary Note 1 (**lines 68-70**):

Molecular formulas were predicted with SIRIUS 4.0 after exporting MS¹ isotope pattern and MS² spectra as a data processing step in MZmine.

Also it is unclear to me if the authors are able to do identify novel adduct types in a targeted or untargeted way. Have they found these novel complexes after identifying the neutral mass/precursor ion or is it embedded in the process?

This is a great question and both the targeted and untargeted discovery of ion species is possible. The different ion annotation algorithms, that are supported by IIMN, use lists of adducts and in-source modifications, e.g, a loss of water. Users can expand this list to cover a wide search space of expected and unexpected ion species. We often do find unexpected adducts using ion identity molecular networking. For example, the $[M+ACN+NH_4]^+$ was unexpected, but can be explained with the addition of NH₄Acetate to an ACN/water gradient. Normally annotations like this require highly trained scientists with good chemical intuition to make sense of it. In this case we could make sense of it by understanding the composition of the chromatography conditions and that it was observed in significant amounts only when ammonia was added. As we included all kinds of ACN in-source clusters, IIMN directly identified these ions and connected them in the resulting networks. In case an ion is missing on the initial list of ion species, there is still a good chance that molecular networking connects this ion with related MS² fragmentation pattern, which enables the manual untargeted discovery of ion species. Ultimately the community, using tools such as IIMN, will develop a list of all possible mass shifts and associate those with the different adducts that are possible. We adjusted the description of how the MZmine workflow, which was used for all datasets in the manuscript, annotates ion identities and handles multiple possible choices in the methods section (**lines 412-420**).

Both the creation and expansion of ion identity networks follow customizable lists of adducts and in-source modifications to cover any type of multimers, in-source fragments, and adducts. The IIN procedure lists all possible ion identity pairs between two features and ranks them according to the

maximum number of features that support a specific annotation, i.e., the ion identity network size. While a feature might be annotated as two different ion species that point to different metabolites, the current workflow will only create additional IIN edges and ion species metadata for the highest-ranking ion identity per feature. This filter decreases the number of spurious matches.

As a summary, I think that the authors have made a lot of work. However, it is not possible from the manuscript description the advantage that this tool represents and its limitations. I think that the authors should be clear about this by providing as many details as necessary to fully understand each one of the choices they are making.

Thanks, this is the only approach that links molecular networks with the different ion forms so that one coherent overview and the relationships of all the MS² spectra in a data set can be better understood. There are many uses for this ability and it is built as part of an analysis ecosystem that sees >200,000 accessions/month. There is a clear need for a better integration of MS¹ and MS² data as others just described a related effort in bioRxiv just a week ago and we now also cite this preprint in the introduction of our paper.² We have aimed to improve the clarity of the tools, including the decision points an end-user has to make to integrate the MS¹ and MS² networks. We have also streamlined the GNPS interface for collapsing ion forms and we have improved the step-by-step online documentation (<https://ccms-ucsd.github.io/GNPSDocumentation/fbmn-iin/>). If there are aspects that still need improvement please let us know the specific passages and we will be happy to address them.

Reviewer #3 (Remarks to the Author):

This paper tackles one of the key problems in the use of untargeted metabolomics: grouping the metabolic peaks from the same compound and merging them in the molecular network for metabolite identification.

Few papers in the past have attempted to solve this problem with different methods. The authors developed Ion Identity Molecular Networking that integrates chromatographic peak shape correlation analysis in the molecular networks to relate ion species of the same metabolites. This paper is unique and interesting by integrating MS1 information and MS2 information and then annotate them in the GNPS tool.

Thanks for the positive comments, this reviewer very much understands the key problem we are addressing with IIMN and the nuances associated with untargeted metabolomics data collection and processing.

I have the following specific comments:

1. Using the IIMN, one feature may match different ‘annotations’ (namely ion identity). For example, feature 1 may be metabolite A with [M+H]⁺ or metabolite B with [M+Na]⁺. So how the IIMN solve this problem?

This is an important question and really depends on the feature finding parameters and algorithm used to annotate ion species. The ion identity molecular networking workflow on GNPS implements interfaces to three popular open source software tools to perform LC-MS feature finding and annotation of ion species. MZmine/metaCorr, XCMS/CAMERA, and MS-DIAL annotate ion species based on different algorithms and validity checks. The new metaCorr algorithm in MZmine annotates all possible combinations and declares the ion species which are supported by the largest ion identity networks (number of related ions) as the best annotations. This means that the algorithm might annotate a feature as two different ion adducts of two different metabolites (A and B). Currently, the export function only considers the best ion identity network for both the ion identity information and the additional IIN edges, which are then used to construct the combined IIMN networks on the GNPS server.

We try to make this behavior more clear in the MZmine part of the methods section (**lines 414-420**):

The IIN procedure lists all possible ion identity pairs between two features and ranks them according to the maximum number of features that support a specific annotation, i.e., the ion identity network size. While a feature might be annotated as two different ion species that point to different metabolites, the current workflow will only create additional IIN edges and ion species metadata for the highest-ranking ion identity per feature. This filter decreases the number of spurious matches.

2. In Figure 1 d and Figure 2 C, the author said that the users can collapse ion identity networks into single molecular nodes to reduce complexity and redundancy. However, I can not find a detailed description of this step. If multiple features in one IIN connect with one feature (MS2 similarity), so after collapsing the IIN, which connection is used in the molecular network?

Good point. We originally performed the collapsing of IIN features as a post processing step in MZmine. However, because we felt this was an important feature of IIMN and due to this question by the reviewer we decided to enable this IIN collapsing step in GNPS before visualizing the results in Cytoscape. By default, GNPS now provides two networking files (.graphml) with and without collapsed IIN as a download. We have added the text below to the MZmine workflow overview. To demonstrate the collapsed IIMN networks from GNPS and new visualization options, we have changed Supplementary Figure 5 (also inserted below). The summed ion intensities can now be visualized as pie-charts on collapsed IIN nodes (b).

To address the changes that were made to the IIMN workflow on GNPS, we added a paragraph to the methods section - Ion identity molecular networking - workflow overview (**lines 360-383**):

1. [...]
9. The option “Download Cytoscape Data” provides two .graphml networking files
 - a. The standard FBMN and IIMN networks (base directory)

- b. IIMN networks with collapsed ion identity networks (in “gnps_molecular_network_iin_collapse_graphml” directory)
10. The option “Direct Cytoscape Preview/Download” provides the IIMN network and its collapsed version as Cytoscape projects with various style presets

Generation of collapsed ion identity networks

One result of the GNPS IIMN workflow is the combined networks with IIN collapsed into single nodes. For this, all ion nodes with the same IIN ID are merged into a representative node based on the feature with the highest library match score, if available, or feature abundance. While all IIN edges are collapsed, MN edges of all ion identities are redirected to their representative nodes so that duplicates replace existing edges if their edge score (cosine similarity) is higher. Limiting the number of MN edges to the one with the highest cosine similarity. Furthermore, representative collapsed nodes are extended by multiple attributes, including the intensity of each ion identity and their summed intensity. This enables the direct comparison of ionization tendencies and provides new visualization options. An example with pie-charts of the ion abundances is demonstrated in Supplementary Figure 5.

Supplementary Figure 5: Ion identity molecular networking results for *Stachybotrys chartarum* liquid culture extracts (MSV000084134). The presented two visualization options compare the combined ion identity molecular network (a), with all ion identities of the same compound sorted in circles, and a collapsed IIN network version (b), with pie-charts representing the relative ion abundances of the corresponding compounds. Compounds A and D were annotated by spectral library match and compounds A, B, D, and E were verified by nuclear magnetic resonance (NMR) spectroscopy²⁰. Modifications between compounds are based on the compound structures or the differences of the average neutral masses for each IIN. The modifications of +O and +H₂O from B to C to F can also be deduced from the ion identity annotations, as the maximum in-source water losses for each compound confirm the addition of oxygens; Compounds A, B, and D (-2H₂O maximum water losses), compound C (-3H₂O), and compound F (-4H₂O). Therefore, this example proves that IIN can yield structure relevant information which facilitates structure annotation and propagation.

3. The authors used the 24 public datasets to validate the IIMN workflow, however, I don't find any analysis about the annotation correct rate of IIMN. So the question is how to know if the annotation results from MMRN are correct or not?

The settings for annotations that were used for MS² library matching typically give rise to FDRs of 1% based on *passatutto*²². Important limitations apply to the identification level depending on the specificity of fragment signals and the precursor *m/z*. Stereoisomers and often structural isomers are indistinguishable by their MS² spectra; hence we are still early in discovering the best FDR strategies (see XY-Meta for example)²⁶.

At this point, we, nor do any of the other papers, do not have an FDR for feature grouping and the identification of ion species and thus are left with manual verification. In light of the comment by this reviewer and users, and to facilitate reviewing, we have now introduced a check if the MS¹ ion identity equals the adduct from an MS² spectral library match and have programmed this to the original network output of the GNPS IIMN workflow (ion identity=spectral library match adduct). The algorithm parses and harmonizes adduct strings before checking for equality. The results are added to the node table as an additional column (see attached Cytoscape screenshot). The shown table is only an excerpt from a bile acid dataset with 46 nodes with both a library match and an ion identity. The new algorithm declares 83% of the nodes to have equal ion species information from both parts of the IIMN workflow. In manual interpretation, however, 7 of the 8 nodes with mismatching ions were identified as related ion adducts with a different number of modifications by -H₂O (mass shift of 18) and the spectral information is insufficient to discriminate those. In one case, we even spotted a user-submitted spectral library entry with incorrect adduct information that we then corrected in the GNPS library. Of course, all libraries have some errors. Earlier, we have even found similar incorrect annotations in the NIST reference library. This kind of measure is valuable for the manual reviewing process.

For the benchmark dataset, Figure 2f shows that the frequency and relative abundance of ion species correspond as expected to the infusion of different salt solutions. These results provide a strong indication that the IIMN workflow with the used MZmine/metaCorrelate algorithm produces a significant correct annotation rate. Furthermore, we experience that the visual exploration of IIMN subnetworks verifies that the new ion identity networking edges connect molecular networks of related compound classes as shown for bile acids and other compounds in Supplementary Figure 4 and 5, respectively.

Best Ion	Adduct	IIN Best Ion=Library Adduct	Compound_Name
[M+Na] ⁺	M+Na	[x]	""(3R,6R)-1-hydroxy-6-((3R,5R,7R,8R,9S,10S,...
[M+Na] ⁺	M+Na	[x]	Spectral Match to Dioctyl phthalate from NIST14
[M+Na] ⁺	M+Na	[x]	""Suspect related to ""Spectral Match to 1-Oc...
[M+NH ₄] ⁺	[M+NH ₄] ⁺	[x]	NCGC00385811-01!6-[3-[(3,4-dimethoxyphenyl...
[M+NH ₄] ⁺	[M+NH ₄] ⁺	[x]	Spectral Match to Taurocholic acid from NIST14 ..
[M+NH ₄] ⁺	[M+NH ₄] ⁺	[x]	Contaminant vial septum ThermoFisher C5000-4.
[M+H] ⁺	[M+H] ⁺	[x]	Spectral Match to Glycocholic acid from NIST14 [.
[M+H] ⁺	[M+H] ⁺	[x]	Spectral Match to Taurocholic acid from NIST14 ..
[M+H] ⁺	[M+H] ⁺	[x]	Spectral Match to Dibutyl phthalate from NIST14.
[M+H] ⁺	M+H	[x]	""2-((4R)-4-((3R,5S,7R,9S,10S,12S,13R,14S,1...
[M+H] ⁺	M+H	[x]	""Suspect related to ""MLS000028461-01!URS.
[3M+H] ⁺	[3M+H] ⁺	[x]	Cholic acid [IIN-based on: CCMSLIB00005435982
[2M+Na] ⁺	2M+Na	[x]	""(4R)-4-((5S,7R,9S,10S,12S,13R,14S,17R)-7,.
[2M+Na] ⁺	2M+Na	[x]	""(R)-4-((3R,5R,8R,9S,10S,12S,13R,14S,17R)-.
[2M+Na] ⁺	2M+Na	[x]	""(R)-4-((3S,5S,7R,8R,9S,10S,13R,14S,17R)-3.
[2M+H] ⁺	[2M+H] ⁺	[x]	Cholic acid [IIN-based: Match]
[2M+H] ⁺	[2M+H] ⁺	[x]	Spectral Match to Taurocholic acid from NIST14 ..
[2M+H] ⁺	2M+H	[x]	""2-((4R)-4-((3R,5R,6R,8S,9S,10R,13R,17R)-3.
[2M+H] ⁺	2M+H	[x]	""(4R)-4-((3R,5R,6S,9S,10R,13R,14S,17R)-3,6.
[2M+H] ⁺	2M+H	[x]	Spectral Match to 13-Docosenamide, (Z)- from N.
[M-H ₂ O+Na] ⁺	M+Na	[ ]	""2-((4R)-4-((5R,7R,8R,9S,10S,12S,13R,17R)-.
[M-H ₂ O+H] ⁺	M+H	[ ]	""Suspect related to ""(R)-4-((3R,5S,7S,8R,9.
[M-H ₂ O+H] ⁺	M+H	[ ]	""(R)-4-((5R,8R,9S,10S,13R,14S,17R)-10,13-d.
[M-H ₂ O+H] ⁺	M+H	[ ]	""2-((4R)-4-((5R,7R,8R,9S,10S,12S,13R,17R)-.
[M-H ₂ O+H] ⁺	M+H	[ ]	Ursodiol
[M-2H ₂ O+H] ⁺	M-H ₂ O+H	[ ]	""2-((4R)-4-((5R,7R,8R,9S,10S,12S,13R,17R)-.

This Cytoscape screenshot of an IIMN node table demonstrates the new column that checks for equality of the „Adduct“ (from spectral library matching) and the „Best Ion“ (from ion identity networking). The test implements a parser that harmonizes different adduct string formats found in the GNPS spectral libraries. This new standard output of the GNPS IIMN workflow facilitates the spotting of discrepancies between MS¹ and MS² annotations.

We have added the following paragraph on the new cross-validation of ion identities and library matches to the methods section in the „Ion identity molecular networking – workflow overview“ (lines 385-406):

Cross-validation of MS² spectral library matches and MS¹ ion identities

In IIMN, nodes may combine annotations from MS² spectral library matching and MS¹ ion identity networking. As cross-validation, GNPS parses and harmonizes the ion species string of both the detected ion identity and matching spectral library entry before checking for equality. The results are reported as an additional column in the node table. This equality check facilitates manual reviewing and the spotting of discrepancies between the MS¹ and MS² annotations.

The ion string parser harmonizes an input (e.g., [M-H₂O+2H]²⁺) in the following steps:

1. Spaces are removed
2. Charge state is detected and removed from the input (2+)
3. Brackets are removed ([])
4. Input is split into added (+2H) and removed (-H₂O) parts

5. Both lists are sorted alphabetically (+2H sorted by letter H)
6. If the charge state is missing, it is calculated for all parts that are listed in a lookup table (e.g., +Na or +H correspond to charge 1+)
7. The harmonized string is constructed by concatenation of [M-all removed parts+all added parts]charge state.

As an example, the harmonized string [M+H]⁺ is produced by the input strings M+H, M+H⁺, and [M+H]⁺, which are all commonly found in the GNPS spectral libraries and as an output of various software tools.

The full open source code of the ion string parser and its latest charge lookup table can be found on GitHub (https://github.com/CCMS-UCSD/GNPS_Workflows).

4. For the MN, we know it is based on MS² similarity, but we know not all the features have MS² spectra, so did those features without MS² spectra be used in IIMN construction? If not, it would lose information for ion identify annotation. If yes, how connect those features with other features in MN?

This is an excellent question, especially as there is a 20-50 fold difference of detectability by MS¹ and triggering of MS² spectra acquisition. Our goal with this workflow is to improve the understanding of MS² based molecular networks. The workflow is built up on feature-based molecular networking which is based on finding MS¹ features across the dataset and picking the representative MS² of the most abundant feature, or merge all MS² to a consensus. We still retain MS¹ feature information where an MS² is not triggered. Hence, it is also possible to apply feature finding and keep all features, even those that are lacking MS² fragmentation data. Consequently, the approach can annotate more ion species without being limited to the most abundant ions with MS² spectra. The IIMN export and direct submission function within MZmine provides both options to either include or exclude features without MS² spectra.

Supplementary figure 5 exemplifies how different ion species add MS¹-derived information to identify and connect compounds. The underlying dataset (MSV000084134) was acquired on an older LTQ-Orbitrap instrument where the gap of MS¹ observation vs triggered MS² spectra was larger than the modern instruments due to slower scan rates. The ion identity networking edges (red, dotted) connect the molecular networks (blue) with more features. The node colors and sizes illustrate that [M+Na]⁺ and in-source fragments of water-losses [M-nH₂O+H]⁺ are more abundant than [M+H]⁺. This illustrates how ion identities (even without MS² spectra) could be used to provide chemical insights and characterization of connected IINs.

Thank you for pointing out that missing part. We have now added details on the MS¹ feature handling to the methods section in the „Ion identity molecular networking – workflow overview“ (lines 341-345).

In comparison to FBMN, IIMN can include features that are lacking MS² fragmentation spectra but are connected to other feature nodes by MS¹ IIN edges. Regarding a higher detectability by MS¹

compared to triggered MS² acquisition, the additional nodes with ion identities complement the resulting networks with information otherwise lost in FBMN or classical MN.

And we tried to clarify the two options, which are provided by the MZmine IIMN export module. This part was added to the methods section in the MZmine workflow description (**lines 422-425**).

The user can limit the export to features with MS² fragmentation spectra or include those with an ion identity. Consequently, the IIMN workflow on GNPS connects features without MS² spectra only by their IIN edges.

5. Not a big deal, but I did not find a detailed description of IIMN workflow in the main text and method section. The “Ion identity molecular networking – workflow overview” in the method section is more like a tutorial of IIMN software. So a detailed description of the IIMN workflow may make the reader easier to understand the concept of the IIMN.

Thanks a lot for bringing this up, while “not a big deal” it is a vital comment. We added a detailed description in a new paragraph in the main text and method section describing the IIMN workflow and emphasizing the focus on LC-MS² data acquired in DDA . We hope that this clarifies the workflow.

The following text was added to the introduction (**lines 133-151**) to better introduce the IIMN workflow and what was done to test it (which also addressed comments by reviewer 1 and 2).

The IIMN workflow annotates and connects related ion species in feature-based molecular networks within the GNPS web platform. We integrate IIN into MS²-based molecular networks and demonstrate the application to LC-MS² studies that make use of product ion scans acquired in data-dependent acquisition (DDA) mode. The IIMN workflow comprises feature grouping, feature shape correlation, and identification of ion species using a variety of feature finding software tools such as MZmine,¹⁰ XCMS,⁹ and MS-DIAL¹¹ that make use of different algorithms for the identification of ion adducts. A table of extracted MS¹ features, each with a consensus MS² spectrum, together with IIN results are then uploaded to GNPS to run the IIMN workflow on the web server. The resulting ion identity molecular networks contain two layers of feature (node) connectivity, linking ion identities of the same compound by MS¹ characteristics and structurally similar compounds by MS² spectral similarity (Figure 1). Here we show an initial validation of IIMN based on a ground truth dataset with induced adduct formation by post-column infusion of salt solutions. Furthermore, we present IIMN results for 2 datasets for natural products standards as well as 24 publicly available experimental datasets. A detailed description of the IIMN workflow as well as a step-by-step tutorial is provided in the method section and can be found online in the GNPS documentation (<https://ccms-ucsd.github.io/GNPSDocumentation/fbmn-iin/>). The IIMN workflow is available online (<https://gnps.ucsd.edu/>) and the source code is shared on GitHub.

New paragraph in the methods section - “Ion identity molecular networking- workflow overview” (lines 316-329):

In general, the ion identity molecular networking (IIMN) workflow starts with LC-MS² data processing in one of the supported open source feature finding tools. After the creation of an aligned feature list of all samples, ion species that originate from the same analyte are grouped and annotated by MS¹ criteria, such as their retention time, feature shape correlation, and m/z difference. Here, such groups are named ion identity networks. Subsequently, information of detected features and their representative MS² spectra, ion identities, and connections to other ion identities are exported and transferred to the GNPS web server for the molecular networking part (refer to tool-specific sections for details). After the construction of ion identity molecular networks, features share connectivity based on MS² spectral cosine similarity and MS¹-based feature shape correlation. In addition to this combined network, GNPS calculates a version with collapsed IIN, where one node represents multiple ions of the same molecule. Results are available in the GNPS web interfaces and as downloads in various open formats as tables and networking files to allow local visualization, reviewing, and post-processing.

References

1. Mahieu, N. G. & Patti, G. J. Systems-Level Annotation of a Metabolomics Data Set Reduces 25 000 Features to Fewer than 1000 Unique Metabolites. *Anal. Chem.* **89**, 10397–10406 (2017).
2. Chen, L. *et al.* Metabolite discovery through global annotation of untargeted metabolomics data. *Cold Spring Harbor Laboratory* 2021.01.06.425569 (2021) doi:10.1101/2021.01.06.425569.
3. Broeckling, C. D., Afsar, F. A., Neumann, S., Ben-Hur, A. & Prenni, J. E. RAMClust: A Novel Feature Clustering Method Enables Spectral-Matching-Based Annotation for Metabolomics Data. *Anal. Chem.* (2014) doi:10.1021/ac501530d.
4. DeFelice, B. C. *et al.* Mass Spectral Feature List Optimizer (MS-FLO): A Tool To Minimize False Positive Peak Reports in Untargeted Liquid Chromatography-Mass Spectroscopy (LC-MS) Data Processing. *Anal. Chem.* **89**, 3250–3255 (2017).
5. Uppal, K., Walker, D. I. & Jones, D. P. xMSannotator: An R Package for Network-Based Annotation of High-Resolution Metabolomics Data. *Anal. Chem.* **89**, 1063–1067 (2017).
6. Kuhl, C., Tautenhahn, R., Böttcher, C., Larson, T. R. & Neumann, S. CAMERA: an integrated strategy for compound spectra extraction and annotation of liquid chromatography/mass spectrometry data sets. *Anal. Chem.* **84**, 283–289 (2012).
7. Senan, O. *et al.* CliqueMS: a computational tool for annotating in-source metabolite ions from LC-MS untargeted metabolomics data based on a coelution similarity network. *Bioinformatics* **35**, 4089–4097 (2019).

8. Jaeger, C., Méret, M., Schmitt, C. A. & Lisec, J. Compound annotation in liquid chromatography/high-resolution mass spectrometry based metabolomics: robust adduct ion determination as a prerequisite to structure prediction in electrospray ionization mass spectra. *Rapid Commun. Mass Spectrom.* **31**, 1261–1266 (2017).
9. Smith, C. A., Want, E. J., O’Maille, G., Abagyan, R. & Siuzdak, G. XCMS: processing mass spectrometry data for metabolite profiling using nonlinear peak alignment, matching, and identification. *Anal. Chem.* **78**, 779–787 (2006).
10. Pluskal, T., Castillo, S., Villar-Briones, A. & Oresic, M. MZmine 2: modular framework for processing, visualizing, and analyzing mass spectrometry-based molecular profile data. *BMC Bioinformatics* **11**, 395 (2010).
11. Tsugawa, H. *et al.* A lipidome atlas in MS-DIAL 4. *Nat. Biotechnol.* **38**, 1159–1163 (2020).
12. Aron, A. *et al.* Native Electrospray-based Metabolomics Enables the Detection of Metal-binding Compounds. doi:10.1101/824888.
13. Frei, A. *et al.* Metal complexes as a promising source for new antibiotics. *Chem. Sci.* **11**, 2627–2639 (2020).
14. Dührkop, K. *et al.* SIRIUS 4: a rapid tool for turning tandem mass spectra into metabolite structure information. *Nat. Methods* **16**, 299–302 (2019).
15. Dührkop, K. *et al.* Systematic classification of unknown metabolites using high-resolution fragmentation mass spectra. *Nat. Biotechnol.* (2020) doi:10.1038/s41587-020-0740-8.
16. Tripathi, A. *et al.* Chemically informed analyses of metabolomics mass spectrometry data with Qemistree. *Nat. Chem. Biol.* **17**, 146–151 (2021).
17. Fraiser-Vannier, O., Chervin, J. & Cabanac, G. MS-CleanR: A feature-filtering approach to improve annotation rate in untargeted LC-MS based metabolomics. *bioRxiv* (2020).
18. Nothias, L.-F. *et al.* Feature-based molecular networking in the GNPS analysis environment. *Nat. Methods* **17**, 905–908 (2020).
19. Aron, A. T. *et al.* Reproducible molecular networking of untargeted mass spectrometry data using GNPS. *Nat. Protoc.* **15**, 1954–1991 (2020).
20. Jagels, A. *et al.* Exploring Secondary Metabolite Profiles of *Stachybotrys* spp. by LC-MS/MS. *Toxins* **11**, (2019).
21. Wang, M. *et al.* Sharing and community curation of mass spectrometry data with Global Natural Products Social Molecular Networking. *Nat. Biotechnol.* **34**, 828–837 (2016).
22. Scheubert, K. *et al.* Significance estimation for large scale metabolomics annotations by spectral matching. *Nat. Commun.* **8**, 1494 (2017).
23. da Silva, R. R. *et al.* Propagating annotations of molecular networks using in silico

- fragmentation. *PLoS Comput. Biol.* **14**, e1006089 (2018).
24. Zhi, H. *et al.* Siderophore-mediated zinc acquisition enhances enterobacterial colonization of the inflamed gut. *Cold Spring Harbor Laboratory* 2020.07.20.212498 (2020)
doi:10.1101/2020.07.20.212498.
 25. Ludwig, M. *et al.* Database-independent molecular formula annotation using Gibbs sampling through ZODIAC. *Nature Machine Intelligence* **2**, 629–641 (2020).
 26. Li, D. *et al.* XY-Meta: A High-Efficiency Search Engine for Large-Scale Metabolome Annotation with Accurate FDR Estimation. *Anal. Chem.* **92**, 5701–5707 (2020).

REVIEWER COMMENTS

Reviewer #2 (Remarks to the Author):

First of all I would like to acknowledge all of the work the authors have made to address my and all of the other reviewers questions. I think that the manuscript has vastly improve. In fact, I am hesitant about the fit of the manuscript in Nature Communications. I can see the use that this tool has to the wider community. However, the way it is explained in the text the tool is a good tool for visualization purposes but it is not apparent to me how this tool advances the state of the art or changes the way in which we think about annotating LC-MS/MS data. I understand that the data-rich visualization approach the authors propose is a first step in the right direction, but I feel that just visualizing in an ordered manner is not enough to warrant publication in Nature Communications. This is in no way my thinking that the work is not of merit (which it is) or sound (which it also is), I think it is a matter of expectations of what I am looking for in an article in Nature Communications.

Let me elaborate further. Using networks for both LC-MS data and MS2 data at the same time is as far as I know new. However, the concept of using networks of similarities between co-elution profiles or MS2 fragmentation spectra is not new (CAMERA, CliqueMS and others use this concept) and the idea of 'connecting' MS2 spectra using cosine similarities between spectra is also not new (see for instance iMet Aguilar-Mogas et al Analytical Chemistry 2017). As I said what is new is considering the two-layers. However, from what I see in the text the two-layer structure is not exploited for annotation. From what I gather in the text, the parental mass identification and collapsed IINN are obtained from LC-MS data (and therefore dependent on the methodology used for that step, correct?). Then MS2 similarities among all of the fragmentation patterns of all of the adducts of each neutral mass are considered to show a summarized visualization. This MS2 data is, as far as I now not used to annotate novel metabolites (or at least it is not described in the text or it was not clear to me).

The authors, in response to one of my comments say that using the two-layer similarity network allows for us to detect/visualize similarities among adducts whose fragmentation spectra are not similar but whose co-elution profiles are. That is true, but the way I understand it, the authors do not exploit this feature. In their workflow, in order to obtain a collapsed IIN they collapse features based on LC-MS data, and then use MS2 data to connect ions with different neutral mass and establish relationships between them. With these steps it looks to me that the fact that two spectra of different adducts are not similar is irrelevant because if they are connected in the LC-MS similarity network they will be collapsed into the same node in the IIN.

The IIN gives the authors obtain a substrate in which they can 'propagate' molecules. Again, this idea is not new: iMet (see previous reference) uses also unitary transformations to establish a network of neutral masses to make predictions of identities of unknown metabolites from the cosine similarities of [M+H] fragmentation spectra. The authors consider similarities of way more adducts and as a result their network is richer, but I think the underlying idea remains the same. Also, I could not find in the text anywhere where the authors show how the tool helps in annotating in any obvious way that I can see since it looks like some of the heavy steps (such as identifying nodes in the collapsed IIN) are done by other tools that are integrated into the workflow and matches are not found using the network (at least from what is explained in the figures and supplementary material).

Other results that the authors show are how they are able to report that in some situation the observe adduct frequencies are different from what one would expect. This is a very interesting finding, but I think that it is not a merit of the workflow but rather the experimental setup. These adducts are as far as I know identified using LC-MS data and not the two-layered network. If it is the case that the authors used the two layers to identify adducts with the same parental mass, I could not find it in the text.

Again I thank very much the authors for their efforts and for answering in detail all of my concerns but I think that this manuscript it a preview of what the future might hold rather than a 'finding', therefore while of merit and a good resource for the community is not a good fit for the journal.

Reviewer #3 (Remarks to the Author):

Thank you for the authors' detailed reply. The responses addressed most of my comments and concerns. I still have several little comments below.

The authors said that "For this, all ion nodes with the same IIN ID are merged into a representative node based on the feature with the highest library match score, if available, or feature abundance." I am a little confused if the feature with the highest library match score and the feature with the most abundance is different, which feature will be selected as the representative node?

I saw that the cosine score is used for MS2 spectra matching, and the feature shape correlation is used for the MS1 feature similarity, I am wondering what are the cutoffs for these criteria and how the author decide them.

Point-to-point reply

Reviewer #2 (Remarks to the Author):

First of all I would like to acknowledge all of the work the authors have made to address my and all of the other reviewers questions. I think that the manuscript has vastly improve.

Thank you very much for acknowledging our work and improvements. Your and the other reviewers' comments definitely helped us to improve in the manuscript and we hope the manuscript is now more clear in describing the method and related advancements in molecular networking.

In fact, I am hesitant about the fit of the manuscript in Nature Communications. I can see the use that this tool has to the wider community. However, the way it is explained in the text the tool is a good tool for visualization purposes but it is not apparent to me how this tool advances the state of the art or changes the way in which we think about annotating LC-MS/MS data. I understand that the data-rich visualization approach the authors propose is a first step in the right direction, but I feel that just visualizing in an ordered manner is not enough to warrant publication in Nature Communications. This is in no way my thinking that the work is not of merit (which it is) or sound (which it also is), I think it is a matter of expectations of what I am looking for in an article in Nature Communications.

Thanks a lot for your comment and for reiterating your concerns. An important point of our approach, which we hope we show more clearly in this revised version of our paper, is the global improvement of annotation and propagation of library matches in non-targeted LC-MS² experiments. We agree with the reviewer that IIMN improves the visualization on MS² datasets (by increased network connectivity). However, the most important point, and we hope we could make this clearer now, is that IIMN overall improves the annotation rate and allows for the generation of new spectral libraries. This is an important contribution to address one of the main bottlenecks in non-targeted metabolomics, which is the limited library coverage and hence low MS² annotation rates.

To elaborate more on this, through propagation of confident library annotations to adjacent ion identities, we created an open access spectral library that shows a broader coverage of ion species. The new IIMN library created in this study from 24 experimental and 2 datasets of authentic standards contains in total 2,657 new entries which have been integrated into the GNPS spectral library. We hope we can convince the reviewer that this serves as a strong example of how IIMN can be further leveraged to expand public MS² libraries in the future. To point this out more clearly and in accordance with the editor's suggestion, we further restructured the manuscript and included an analysis of the coverage and distribution of ion identities in public libraries (previously found in the SI) in the main text as Figure 6.

In addition, we created a new figure (main text, Figure 4) showcasing the increase in annotations, through IIMN propagation of first network neighbors in the 24 experimental datasets. On average, we reached an increase of 35% of the MS² annotation rate through IIMN, which in our experience is a significant improvement. We added the following paragraph to describe this in the main text **(lines 212-227)**:

From a global view on all 24 datasets, IIMN successfully reduced the number of unconnected LC-MS² features and increased the connections to annotated compound structures (Supplementary Figure 6, Supplementary Table 2). Annotation rates in all 24 datasets of 6% and 12% are in the expected range with contemporary MS² library matching^{1,2} and MS¹ ion annotation, respectively, especially with the here chosen restrictive IIN parameters (Figure 4a). By propagating spectral library matches to first neighboring IIN nodes, the annotation rates of the test datasets were increased by an average of 35% (Figure 4a,b). On the individual dataset level, the highest increase (325%) was observed for dataset 4 with more MS¹ data points per feature and thus better feature shape correlation on the cost of a lower MS² acquisition rate. Most datasets (16 out of 24) experienced an increase greater than 10%, while eight datasets were below this value. After inspecting the LC-MS² files, we found various reasons for this difference. Datasets 11 and 12, for example, had a higher focus on MS² acquisition with a high topN of MS² events in the DDA settings that caused lower MS¹ survey scan frequencies and hence fewer data points per features, resulting in lower IIN correlation and connectivity. For datasets 7 and 19, the MS² annotation rate was low to begin with and hence few annotations could be propagated by IIMN.

Review Figure 1 (Main Text Figure 4): Overview of IIMN results for 24 experimental datasets. a) Summarizes the relative number of LC-MS features (with MS² spectrum) that were annotated by ion identities or matches to the GNPS spectral libraries. The increased annotation rate by propagating library matches to connected unannotated ion identities is highlighted and **b)** displayed as relative gains with a mean increase by 35% compared to all library matches. **c)** Comparison of relative ion formation tendencies measured as the number of ion identities.

Besides the propagation of level 2 annotations (based on accurate mass and spectral similarity), IIMN results in an average ion species annotation rate of 12% for the 24 test datasets, which can be further leveraged for molecular formula calculation and in-silico structure prediction, e.g., with the SIRIUS software. This is mentioned in the main text and made available through interfaces to other community projects (lines 266-271):

The integration into FBMN and the GNPS environment provided a platform to utilize IIMN in other related bioinformatics tools, e.g., SIRIUS³, CANOPUS⁴, and Qemistree⁵ for molecular formula and compound class level annotation, which will also facilitate additional validation of network connectivity. Direct interfaces to the GNPS-Dashboard and MASST⁶ support collaborative data visualization and repository scale MS² queries, respectively.

Let me elaborate further. Using networks for both LC-MS data and MS² data at the same time is as far as I know new. However, the concept of using networks of similarities between co-elution profiles or MS² fragmentation spectra is not new (CAMERA, CliqueMS and others use this concept) and the idea of 'connecting' MS² spectra using cosine similarities between spectra is also not new (see for instance iMet Aguilar-Mogas et al Analytical Chemistry 2017).

As I said what is new is considering the two-layers. However, from what I see in the text the two-layer structure is not exploited for annotation. From what I gather in the text, the parental mass identification and collapsed IINN are obtained from LC-MS data (and therefore dependent on the methodology used for that step, correct?). Then MS² similarities among all of the fragmentation patterns of all of the adducts of each neutral mass are considered to show a summarized visualization. This MS² data is, as far as I now not used to annotate novel metabolites (or at least it is not described in the text or it was not clear to me).

We agree with the reviewer, and we believe this is what we describe in the main text. We highlight that both CAMERA and CliqueMS can be used for networking on MS¹ data, and CAMERA, for example, can be used as a correlation module within IIMN (with XCMS). Further, we described that IIMN builds up on FBMN, which is based on cosine MS² similarity. We believe that this method is applied by other tools underlines its importance in mass spectrometry research. Here we also integrate both tools, IIMN is the only tool that integrates both and we provide the ability for the rest of the scientific community to do the same through step-by-step instructions and access to computational infrastructure. Further, the merging of both methods will accelerate research and the development of even newer algorithms that use both pieces of information.

We appreciate the reviewer mentioning the iMET tool. As many tools are being developed that help the community, we try to provide a platform that integrates many approaches with a data repository and community driven resources. iMET seems to be a useful

contribution based on reading the paper. The tool, however, does not appear to be currently maintained. We reached out to the developers of iMET after failing to process the test data given on their web server. According to them, the source code is closed and not available. One of the authors mentioned they have “something better in the pipeline”.

A crucial part in developing new tools is to make them accessible for the broad community. Our IIMN workflow is open source, extendible, and integrated into a comprehensive analysis platform, which provides further options for downstream analysis. IIMN implements interfaces to other community tools, e.g., SIRIUS and Qemistree for annotations, the GNPS-Dashboard for collaborative data visualization, and MASST for repository scale MS² spectrum search to mention a few. These tools now get 300,000 accessions a month.

In the conclusion, we emphasized the importance of an open web-platform and source code for the integration of other community projects and to making analysis tools available to other researchers **(lines 266-274)**:

The integration into FBMN and the GNPS environment provided a platform to utilize IIMN in other related bioinformatics tools, e.g., SIRIUS³, CANOPUS⁴, and Qemistree⁵ for molecular formula and compound class level annotation, which will also facilitate additional validation of network connectivity. Direct interfaces to the GNPS-Dashboard and MASST⁶ support collaborative data visualization and repository scale MS² queries, respectively. Furthermore, the open source code and generic connection between feature finding, ion identity molecular networking, and the online GNPS workflow encourage the implementation of interfaces to other feature grouping and ion identification algorithms.

The authors, in response to one of my comments say that using the two-layer similarity network allows for us to detect/visualize similarities among adducts whose fragmentation spectra are not similar but whose co-elution profiles are. That is true, but the way I understand it, the authors do not exploit this feature. In their workflow, in order to obtain a collapsed IIN they collapse features based on LC-MS data, and then use MS² data to connect ions with different neutral mass and establish relationships between them. With these steps it looks to me that the fact that two spectra of different adducts are not similar is irrelevant because if they are connected in the LC-MS similarity network they will be collapsed into the same node in the IIN.

Thanks a lot for the comment. If we understand the reviewer correctly, then this perfectly describes the drawbacks of FBMN that we try to solve with IIMN. In classical MN and FBMN, that is used by researchers all over the world to annotate and propagate related MS² spectra, different ion species create redundancy and result in dispersed unconnected subnetworks and nodes. IIMN now leverages LC-MS feature shape correlation and other filters to successfully connect ion species of the same molecules (as shown in Figures 2 and 5 and Supplementary Figure 4). IIMN in GNPS is the first tool that connects the two-layer

similarity. The optional collapsing of ion identity networks into single molecular nodes reduces the complexity and is performed after creating all connections. Both the full and collapsed networks are provided in open formats and can be further analyzed either visually or computationally. This creates the option to traverse the network nodes and use all metadata and the results from IIMN and related tools for further processing.

The IIN gives the authors obtain a substrate in which they can 'propagate' molecules. Again, this idea is not new: iMet (see previous reference) uses also unitary transformations to establish a network of neutral masses to make predictions of identities of unknown metabolites from the cosine similarities of [M+H] fragmentation spectra. The authors consider similarities of way more adducts and as a result their network is richer, but I think the underlying idea remains the same. Also, I could not find in the text anywhere where the authors show how the tool helps in annotating in any obvious way that I can see since it looks like some of the heavy steps (such as identifying nodes in the collapsed IIN) are done by other tools that are integrated into the workflow and matches are not found using the network (at least from what is explained in the figures and supplementary material).

We agree with the reviewer that the concept of propagating annotation is not new. We acknowledged previous works as this is exactly what molecular networking in GNPS is designed to do^{7,8}. However, as pointed out in the manuscript, the main bottleneck of this propagation was that MN often fails to connect different ion species based on their MS² spectra. This is the fundamental and key challenge IIMN is solving. It is the most asked question about molecular networking - How do we connect different adducts, in source fragments etc.? IIMN overcomes this fundamental problem that plagues every LC-MS² metabolomics study by combining MS² and MS¹ correlation networks (what the reviewer refers to as a two layer network).

In our opinion, in addition to providing a solution to a very fundamental and common problem, one of the main advances by IIMN is its availability in widely used bioinformatic data analysis pipelines with MZmine, XCMS, or MS-DIAL directly linked to IIMN on GNPS. IIMN can easily be integrated into standard untargeted LC-MS² processing workflows.

Finally, IIMN also improves annotation rates. We added the above mentioned Figure 4 and a corresponding new paragraph (see above) that highlights the increase in annotation rates in comparison to standard FBMN, which led to the generation of IIMN-based spectral libraries of 2,657 propagated library entries. We updated the description of the potential of IIMN for new spectral libraries in the main text **(lines 228-230)**:

Besides the increase in feature annotations in individual datasets, IIMN also enables the generation of new propagated spectral libraries, increasing and diversifying the library coverage beyond commonly considered ion species.

[..] (lines 241-246)

Here, IIMN can be used to expand the spectral libraries with additional adducts and in-source fragments in LC-MS experiments, which can significantly increase spectral library coverage and thus MS² annotation rates. The potential to use IIMN to propagate spectral library matches to adjacent unannotated features with ion identity is evident from a mean increase of the annotation rate by 35% (Figure 4a,b).

Other results that the authors show are how they are able to report that in some situation the observed adduct frequencies are different from what one would expect. This is a very interesting finding, but I think that it is not a merit of the workflow but rather the experimental setup. These adducts are as far as I know identified using LC-MS data and not the two-layered network. If it is the case that the authors used the two layers to identify adducts with the same parental mass, I could not find it in the text.

Thanks a lot for the comments and we agree with the reviewer that IIMN enables a new perspective on adduct frequencies which we anticipate will reduce feature redundancy and increase annotation rates and library diversification. The reviewer is also correct that IIMN annotates adducts based on mass deltas of the MS¹ correlation network. The two-layered network was then used to propagate MS² library annotations to adducts and in-source fragments of matched compounds. Here, we created a new spectral library with 2,657 library entries that were propagated with a conservative parameter selection. All experimental datasets were acquired with varying LC-MS conditions on different instruments in DDA mode. The concept of library entry propagation is described in the methods section in the "IIMN-based spectral library generation" subsection (lines 642-698).

Again I thank very much the authors for their efforts and for answering in detail all of my concerns but I think that this manuscript is a preview of what the future might hold rather than a 'finding', therefore while of merit and a good resource for the community is not a good fit for the journal.

Thanks again for the reviewer's time and feedback. We are glad to have the opportunity to address in detail their concerns. IIMN is available in MZmine, XCMS, and MS-DIAL together with GNPS, so while we hope it will be further improved in the future, all our tools continue to be improved. This paper is not a preview, IIMN is not a script deposited in Github that only the developers can use. The computational infrastructure and step-by-step documentation are available to the entire community as long as they have access to a computer connected to the web. In addition, we also organized an online workshop to assist the community interested in using this tool. Here we not only offer the first solution that integrates two networks but we also provide the infrastructure for the larger scientific community to do so - this is 95% of the entire effort as we did not just want to have a "proof-of-principle" demonstration but rather something functional and stable for others to use. Our paper and accompanying documentation is not a preview, but rather a presentation and validation of what is available today for the entire scientific community.

IIMN has become an integral part of the standard data analysis pipeline for non-targeted metabolomics, molecular networking, and library expansion in the MZmine and GNPS analysis ecosystems. This is consistent with the impact of a paper in Nature Communications.

Reviewer #3 (Remarks to the Author):

Thank you for the authors' detailed reply. The responses addressed most of my comments and concerns. I still have several little comments below.

Thanks again for the detailed feedback and cautious reading of the manuscript. We appreciate your investment in helping us improve it.

The authors said that "For this, all ion nodes with the same IIN ID are merged into a representative node based on the feature with the highest library match score, if available, or feature abundance." I am a little confused if the feature with the highest library match score and the feature with the most abundance is different, which feature will be selected as the representative node?

Thanks for pointing us to this ambiguity. The actual behavior is to merge all nodes of the same IIN into the feature with the highest library match score. Only if there is no library match in an IIN, the feature with the maximum abundance is used as the base for merging. We have changed the sentence in the following way to clarify the statement.

For this, all ion nodes with the same IIN ID are merged into a representative node based on the feature with the highest library match score, if available, or otherwise the feature with the maximum abundance.

I saw that the cosine score is used for MS2 spectra matching, and the feature shape correlation is used for the MS1 feature similarity, I am wondering what are the cutoffs for these criteria and how the author decide them.

Thanks a lot, this is a great question. In the modified cosine similarity-based molecular networking, we typically use parameters for spectral library matching that correspond to an estimated false discovery rate (FDR) of 1%^{8,9} as a starting point. For the analysis of 24 datasets, co-authors used their dataset specific MN parameters for each of the different projects, which are listed in Supplementary Table 6. For the MS¹ feature shape correlation, we used the metaCorrelate algorithms in MZmine with mostly default parameters (>50% intensity overlap, feature shape Pearson correlation greater equals 0.85 with at least 5 data points and 2 data points on each edge). The optimal parameters depend on the LC-MS instrument, method, study design, and goal in mind. Some projects use very high MS² acquisition rates, which often have a negative impact on the MS¹ feature shape and feature

detection in general (e.g., Figure 4, datasets 11 and 12). Then users might want to use less restrictive feature grouping parameters. On the other hand, a higher MS¹ acquisition rate might lead to decreased MS² coverage (e.g., Figure 4, dataset 4).

We describe the feature correlation and ion identity networking steps in the methods section under “IIMN with MZmine” in **lines 499-539**. An excerpt is provided below (**lines 502-508**):

The feature shape correlation is a vital filter to reduce false grouping significantly and can apply either a minimum Pearson correlation (favored) or cosine similarity. A requirement is at least five data points, two on each side of the peak apex. If a low MS¹ scan rate leads to chromatographic peaks with less than five data points, it is advisable to either redesign the acquisition method or to turn off the feature shape correlation. Note that the latter is expected to reduce the ion annotation consistency and should be used with caution.

As this is related to the data quality and acquisition method, we now describe some reasons why the experimental datasets result in different MS² and MS¹ annotation rates and potentials for propagation of annotations to first IIMN neighbors, in the main text (**lines 219-227**):

On the individual dataset level, the highest increase (325%) was observed for dataset 4 with more MS¹ data points per feature and thus better feature shape correlation on the cost of a lower MS² acquisition rate. Most datasets (16 out of 24) experienced an increase greater than 10%, while eight datasets were below this value. After inspecting the LC-MS² files, we found various reasons for this difference. Datasets 11 and 12, for example, had a higher focus on MS² acquisition with a high topN of MS² events in the DDA settings that caused lower MS¹ survey scan frequencies and hence fewer data points per features, resulting in lower IIN correlation and connectivity. For datasets 7 and 19, the MS² annotation rate was low to begin with and hence few annotations could be propagated by IIMN.

References

1. Petras, D. *et al.* High-Resolution Liquid Chromatography Tandem Mass Spectrometry Enables Large Scale Molecular Characterization of Dissolved Organic Matter. *Frontiers in Marine Science* **4**, 405 (2017).
2. Gauglitz, J. M. *et al.* Untargeted mass spectrometry-based metabolomics approach unveils molecular changes in raw and processed foods and beverages. *Food Chem.* **302**, 125290 (2020).
3. Dührkop, K. *et al.* SIRIUS 4: a rapid tool for turning tandem mass spectra into metabolite structure information. *Nat. Methods* **16**, 299–302 (2019).
4. Dührkop, K. *et al.* Systematic classification of unknown metabolites using high-resolution fragmentation mass spectra. *Nat. Biotechnol.* (2020) doi:10.1038/s41587-020-0740-8.
5. Tripathi, A. *et al.* Chemically informed analyses of metabolomics mass spectrometry data with Qemistree. *Nat. Chem. Biol.* **17**, 146–151 (2021).
6. Wang, M. *et al.* Mass spectrometry searches using MASST. *Nat. Biotechnol.* **38**, 23–26 (2020).
7. Watrous, J. *et al.* Mass spectral molecular networking of living microbial colonies. *Proc. Natl. Acad. Sci. U. S. A.* **109**, E1743–52 (2012).
8. Wang, M. *et al.* Sharing and community curation of mass spectrometry data with Global Natural Products Social Molecular Networking. *Nat. Biotechnol.* **34**, 828–837 (2016).
9. Scheubert, K. *et al.* Significance estimation for large scale metabolomics annotations by spectral matching. *Nat. Commun.* **8**, 1494 (2017).